# A two-step actin-mediated strategy enables *Campylobacter jejuni* to promote mitochondrial aggregation and iron homeostasis, for intracellular survival and persistence
Fauzy Nasher ⓘ ✉ & Brendan W. Wren ⓘ ✉

*Campylobacter jejuni*, a major cause of bacterial gastroenteritis, is capable of surviving in diverse hosts, including free-living amoebae such as Acanthamoeba. However, the molecular mechanisms that facilitate its intracellular persistence and subsequent transfer remain poorly defined. Here, we hypothesize that *C. jejuni* employs a biphasic actin-remodelling strategy, mediated by the effector proteins CiaI and CiaD, to reposition and remodel host mitochondria, promoting mitochondrial aggregation and iron homoeostasis. Using dual proteomics, microscopy, biochemical assays, and defined genetic mutants, we show that actin polymerization and CiaI are critical for mitochondrial interaction. We found that CiaI binds nucleotides with cooperative kinetics, acting as a molecular switch, and is crucial for *C. jejuni* localization near mitochondria, while CiaD promotes actin polymerization and acanthopodia formation to facilitate uptake. We propose a two-phase model: early actin polymerization repositions mitochondria, followed by localized actin depolymerization and mitochondrial remodelling. Iron chelation promotes bacterial survival, suggesting that oxidative stress functions as a host defence. These findings highlight a sophisticated mechanism of intracellular adaptation by *C. jejuni* that may be relevant to pathogenesis and identify new potential targets for disrupting its environmental and clinical persistence.

Pathogenic bacteria have evolved mechanisms to exploit host cellular processes, often targeting organelles like mitochondria to modulate stress responses and enhance survival[1,2]. These organelles, believed to have evolved from free-living bacteria[3], possess sophisticated innate defence mechanisms, including the generation of reactive oxygen species (ROS) such as hydrogen peroxide ($H_2O_2$), which act as potent antimicrobial agents[4,5]. However, many pathogens have evolved strategies to subvert mitochondrial defence mechanisms, influencing host cell fate and promoting their own survival[6,7]. Recent studies have shown that bacteria manipulate mitochondrial dynamics, altering processes like fusion and fission to evade immune responses and gain a survival advantage in both protozoan and mammalian hosts[2,8–10].

Bacteria often exploit mitochondrial bioenergetics by altering ATP production to redirect energy towards their persistence while suppressing host immune defences[7]. Pathogens also disrupt mitochondrial calcium homoeostasis, interfering with critical intracellular signalling pathways that regulate apoptosis and inflammation[11]. Another strategy involves manipulating mitochondrial iron metabolism, which is crucial for both host cell function and bacterial growth, by altering iron availability to support their own metabolic requirements[12]. *Salmonella enterica* manipulates host cell mitochondria by interfering with ROS production by delaying mitochondria recruitment to the bacterial vacuole and prevent apoptosis by inhibiting cytochrome c release[13]. These strategies demonstrate how bacteria manipulate mitochondrial functions not only for survival but also to actively modulate host cell processes, thereby enhancing pathogenic potential.

*Campylobacter jejuni* is the leading cause of bacterial gastroenteritis worldwide[14], yet its ability to persist and disseminate in various

Department of Infection Biology, London School of Hygiene and Tropical MedicineKeppel St, London, UK.
✉e-mail: fauzy.nasher1@lshtm.ac.uk; brendan.wren@lshtm.ac.uk

environments remains poorly understood. Unlike some Gram-negative pathogens, *C. jejuni* does not possess the classical virulence factors commonly found in other bacteria, including the canonical type III secretion system (T3SS)[15]. The classical T3SS, found in many pathogenic Gram-negative bacteria, shares structural similarities with the bacterial flagellum[16]. It was later discovered that this flagellar T3SS could also export non-flagellar proteins, including virulence factors[17]. In *C. jejuni*, the flagellar T3SS (fT3SS) has been shown to secrete non-flagellar proteins[18], such as the Campylobacter invasion antigens (Cia proteins), particularly in response to contact with host cells or host-derived components[19]. To date, four Cia proteins have been identified: CiaB (Cj0914c), CiaC (Cj1242), CiaD (Cj0788) and CiaI (Cj1450). Current evidence suggests that these Cia proteins hijack elements of the host focal complex—a cell-matrix adhesion structure—and activate the MEK/ERK signalling pathway. This interaction leads to membrane ruffling, including the formation of lamellipodia and filopodia, which are actin-supported cellular protrusions that facilitate *C. jejuni* invasion into host cells[20].

While traditionally considered a warm-blooded host-adapted pathogen, emerging evidence suggests that *C. jejuni* can exploit free-living amoebae (FLA) as environmental reservoirs[21,22]. FLA, such as Acanthamoeba, was shown to enhance the survival of *C. jejuni* outside warm-blooded hosts and its transmission[23–25]. Acanthamoeba serve as training grounds for bacterial pathogens[26], including *C. jejuni*[23], providing a niche that offers protection from environmental stressors and facilitating bacterial adaptation to host immune responses. Recent studies demonstrate that *C. jejuni* modulates key amoeba cellular processes to its advantage, including interaction with host mitochondria. Similarly, *C. jejuni* interacts with mitochondria not only in amoebae but also in human colonic epithelial T84 cells, indicating its adaptability across diverse host types[27]. Consistently, lysates of *C. jejuni* can depolarize mitochondria in HeLa cells, reinforcing the importance of bacterial-mitochondrial interactions in pathogenesis[28].

In this study, we investigate the interaction between *C. jejuni* and *A. castellanii* mitochondria to understand how the bacterium exploits the amoeba's intracellular environment. Using dual proteomics, fluorescence microscopy, survival assays, and biochemical analyses, we demonstrate that *C. jejuni* alters both its own proteome and the host mitochondrial profile within *A. castellanii*. Our findings reveal that *C. jejuni* modulates actin dynamics through Campylobacter invasion antigens, CiaD and CiaI, a key strategy for persistence in amoebae. Additionally, we show that *C. jejuni* infection induces mitochondrial iron redistribution, which may influence oxidative stress and enhance cytochrome c oxidase activity, optimizing ATP production while mitigating ROS damage.

Building on these insights, we propose a two-phase model in which *C. jejuni* first employs CiaD to promote actin polymerization and reposition mitochondria near the bacterial phagosome. In the later phase, CiaI induces localized actin depolymerization, remodelling mitochondria to further enhance bacterial survival. This sequential strategy ensures that *C. jejuni* can establish intracellular positioning before fine-tuning its host environment, highlighting a sophisticated mechanism of host manipulation.

These findings enhance our understanding of *C. jejuni* adaptation and pathogenesis, extending bacterial survival mechanisms beyond mammalian hosts. They also reveal the role of protozoa in *C. jejuni* ecology, highlighting potential targets to disrupt its persistence in environmental settings and potential drug targets for the treatment of Campylobacteriosis.

## Results
### Dual proteomics reveal host cytoskeletal and mitochondrial remodelling
We previously reported an intimate interaction of *Campylobacter jejuni* strain 11168H with fused mitochondria of both *Acanthamoebae castellanii* and T84, a human colonic epithelial cell line[27]. To investigate how *C. jejuni* modulates host mitochondria during infection, we isolated mitochondria-enriched fractions from *A. castellanii* infected with *C. jejuni* 11168H after

4 h and subjected these to data-independent acquisition (DIA) -based liquid chromatography mass spectrometry (LC-MS/MS) proteomic analysis. This approach provided a spatially resolved view of proteins associated with or enriched near mitochondria during infection, rather than a global expression profile. We detected high-confidence proteins from both host and pathogen (Quant Score $\geq 20$, $p < 0.01$). Quantitative values reflect interference-free protein-level intensity (ifq.dia.protein), ensuring accurate representation of protein presence (Table 1).

LC–MS/MS analyses identified a range of proteins from both *A. castellanii* and *C. jejuni* 11168H, offering insights into the molecular interactions between these organisms. Table 1 lists all host and bacterial proteins detected above our statistical confidence threshold (Quant Score $\geq 20$; $p < 0.01$) within the mitochondrial-enriched fractions, representing the complete set of significantly detected proteins rather than a selected subset. Full mass spectrometry datasets are provided in Supplementary Data 1 (all quantified proteins) and Supplementary Data 2 (proteins meeting a Quantification Score $\geq 20$). The raw and processed data are also publicly available via ProteomeXchange under the identifier PXD065850.

Proteins detected in *A. castellanii* include actin and actin-related proteins (P02578, P37167, P53487, P53490) and Actin-1, the core component of cytoskeleton[29]. Key regulators of actin branching, Arp2 and Arp3 important for complex-mediated actin branching[30] and Actophorin, an ADF/cofilin homologue that depolymerizes actin filament[31] were also confidently identified. Multiple myosin family members (P05659, P19706, P10569) key intracellular trafficking including mitochondrial positioning, phagosome transport and membrane dynamics[32] were detected alongside profilin family members (P68696, Q95VF7, P19984), which regulate actin polymerization by promoting ATP-actin incorporation[33].

Mitochondrial ATP synthase subunits (Q37380, Q37377) and cytochrome c oxidase components (Q37370, Q37374)—essential elements of the electron transport chain (ETC) for ATP production and electron transfer[34], were also prominently identified in *A. castellanii* infected with *C. jejuni*. Additionally, the vacuolar iron transporter Ccc1/VIT1 (L8GFE0), which supports mitochondrial iron homoeostasis by mediating iron sequestration into vacuoles[35], was also detected. Although individual amoeba proteins are not one-to-one orthologs of their mammalian counterparts, their conserved domains and biochemical functions are largely maintained. Overall, this supports the notion that *C. jejuni* may interact with analogous host pathways in both amoebae and mammalian cells and exploit common eukaryotic mechanisms despite evolutionary divergence.

In parallel, *C. jejuni* proteins identified in the infected mitochondrial fractions included PorA (P80672), a major outer membrane protein involved in nutrient uptake and host cell interaction[36]. Several proteins associated with bacterial adaptation and metabolism were also present, such as TsaD (A8FN22), a tRNA-modifying enzyme required for translational fidelity; Mdh (Q9PHY2), malate dehydrogenase, a TCA cycle enzyme[37]; and AcpP (A1VYF9), an acyl carrier protein essential for fatty acid biosynthesis. Stress response proteins, including KatA (Q59296)[38], thioredoxin peroxidase, Tpx (Q9PPE0), DNA-binding protein/bacterioferritin, Dps/Bfr (Q0P891)[39], the periplasmic methionine sulfoxide reductase, MsrP (Q9PIC3), and chaperonins GroES (Q5HTP3) and GroEL (A7H2F8)[40], were also identified, suggesting bacterial oxidative stress adaptation within the host environment. The detection of flagellar proteins FlaA and FlaB, (P56963 and P56964, respectively) further emphasise the importance of flagella in host-pathogen interactions[41,42]. Notably, *C. jejuni* flagella are heavily glycosylated and play a unique dual role in motility and the secretion of host-targeting effectors, including Cia proteins[20].

Based on the prominent detection of cytoskeletal proteins, particularly actin and actin-related proteins, we hypothesize that actin dynamics play a crucial role in the association of *C. jejuni* with mitochondria during infection.

**Table 1 | Proteomics identification of host and bacterial proteins enriched in *A. castellanii* infected with *C. jejuni* 11168H**

| Accession | Gene | -10LogP | Description |
|---|---|---|---|
| *Acanthamoeba castellanii* | | | |
| P02578 | | 183.48 | Actin-1 |
| P05659 | | 141.21 | Myosin-2 heavy chain, non-muscle |
| Q37380 | ATP1 | 117.27 | ATP synthase subunit alpha, mitochondrial |
| Q37370 | COX1/2 | 102.28 | Cytochrome c oxidase subunit 1 + 2 |
| Q95032 | metK | 98.85 | S-adenosylmethionine synthase |
| P53490 | ARP3 | 92.31 | Actin-related protein 3 |
| P19706 | MIB | 87.26 | Myosin heavy chain IB |
| P49633 | | 85.54 | Ubiquitin-ribosomal protein eL40 fusion protein |
| P90513 | | 79.43 | Proteasome subunit alpha type-3 (Fragment) |
| P10569 | MIC | 75.47 | Myosin IC heavy chain |
| Q37377 | ATP9 | 70.61 | ATP synthase subunit 9, mitochondrial |
| P68696 | | 67.61 | Profilin-1A |
| P37167 | | 63.35 | Actophorin |
| P53487 | ARP2 | 59.39 | Actin-related protein 2 |
| Q95VF7 | | 58.17 | Profilin-1B |
| P19984 | | 49.98 | Profilin-2 |
| Q37374 | COX3 | 42.23 | Cytochrome c oxidase subunit 3 |
| L8GFE0 | VIT | 39.41 | Vacuolar iron transporter |
| Q9TGM3 | | 30.37 | Uncharacterized mitochondrial protein Mp36 |
| *Campylobacter jejuni* 11168H | | | |
| P80672 | porA | 84.61 | Major outer membrane protein |
| Q0PC30 | atpD | 50.08 | ATP synthase subunit beta |
| P56963 | flaA | 48.13 | Flagellin A |
| Q0P891 | dps/bfr | 44.74 | DNA protection during starvation protein |
| A8FN22 | tsaD | 40.00 | tRNA N6-adenosine threonylcarbamoyltransferase |
| P56964 | flaB | 39.06 | Flagellin B |
| Q9PHY2 | mdh | 33.53 | Probable malate dehydrogenase |
| Q5HTP3 | groES | 29.15 | Co-chaperonin |
| A7H361 | cmoB | 24.70 | tRNA U34 carboxymethyltransferase |
| A7H2F8 | groEL | 23.29 | Chaperonin |
| Q46106 | cft | 21.02 | Bacterial non-haem ferritin |
| Q9PPE0 | tpx | 19.15 | Thiol peroxidase |
| A1VYF9 | acpP | 18.32 | Acyl carrier protein |
| Q59296 | katA | 17.31 | Catalase |
| Q9PIC3 | msrP | 15.94 | Protein-methionine-sulfoxide reductase catalytic subunit |

Data is presented as $^{-10}$LogP value > 20 = $p < 0.01$. Data represents three independent biological replicates.

### *C. jejuni* interaction with *A. castellanii* mitochondria is actin-dependent

To investigate whether actin polymerization is required for *C. jejuni*-mitochondria interactions, we treated *A. castellanii* with actin inhibitors (cytochalasin D and CK-666) and assessed *C. jejuni* localization relative to mitochondria after 4 h p.i (Fig. 1). Untreated control showed *C. jejuni* associates with *A. castellanii* mitochondria, with a strong presence of actin filaments (Fig. 1a).

*A. castellanii* was treated with actin polymerization inhibitor cytochalasin D (10 µm) and as expected, actin filaments were significantly reduced, which was accompanied by a marked decrease in *C. jejuni*-mitochondrial association (Fig. 1b). These findings indicate that actin polymerization is critical for *C. jejuni* interaction with mitochondria.

Given the high confidence detection of Arp2 and Arp3 detected in our proteomics data (-10LogP of 92.31 and 59.39, respectively), we hypothesized that the Arp2/3 complex, which promotes the nucleation of branched actin filament networks[43], facilitates *C. jejuni*-induced actin remodelling. Arp2/3 complex activity was inhibited using CK-666 (50 µm) and C. jejuni localization was examined relative to mitochondria. Arp2/3 inhibition resulted in a substantial reduction in C. jejuni association with *A. castellanii* mitochondria (Fig. 1c), alongside a corresponding decrease in actin filaments, similar to the effect observed with cytochalasin D (Fig. 1d). It is noticeable that inhibition of myosin II with blebbistatin did not hinder *C. jejuni* association with *A. castellanii* mitochondria, emphasizing the direct role of actin during host-pathogen interaction (Supplementary Fig. 1a). Quantification of mitochondrial aggregates confirmed the qualitative observations: close association of *C. jejuni* with host mitochondria showed significantly larger mean mitochondrial object size ($p < 0.001$), consistent with the formation of mitochondrial aggregates, whereas treatment with cytochalasin D or CK-666 shifted mitochondria towards a more punctate morphology with a reduced mean object size (Fig. 1e). Mean fluorescent intensity (MFI) of phalloidin-stained actin also confirmed these effects: cytochalasin D and CK-666 significantly reduced actin MFI compared with both untreated infected and uninfected controls ($p < 0.001$ for both). DMSO (vehicle control) alone did not visibly alter *C. jejuni*-induced mitochondrial aggregation or actin organisation; DMSO-treated infected amoebae showed mitochondrial morphology and actin filament patterns comparable to untreated infected cells (Supplementary Fig. 1b).

We previously observed statistically significant ($p < 0.05$) over-expression of genes encoding Cia proteins, including *ciaB* (*Cj0914c*), *ciaC* (*Cj1242*), and *ciaI* (*Cj1450*) in our intra-amoebae *C. jejuni* RNA-Seq analyses[44]. *C. jejuni* 11168HΔ*ciaB* mutant failed to associate with mitochondria and exhibited reduced intracellular survival[27]. CiaB is essential for the secretion of other Cia effectors[18]. In this study, we examined CiaC, CiaI and CiaD effectors, which are actively translocated into host cells[20] and may influence mitochondrial dynamics. Notably, the secretion of Cia proteins is mediated by the fT3SS, which is intricately linked to proper flagellar assembly[45,46]. Both FlaA and FlaB were abundantly expressed in our proteome analyses, indicating that the flagellar system is not only present but likely active.

### CiaI promotes localized actin remodelling and mitochondrial aggregation

We examined the ability of *C. jejuni* 11168HΔ*ciaC* and Δ*ciaI* mutants to associate with *A. castellanii* mitochondria. The 11168HΔ*ciaC* mutant retained its ability to interact with mitochondria, and its intra-amoeba survival was indistinguishable from the wild-type strain after 4 h intracellular survival (Supplementary Fig. 2). This suggests that CiaC is not required for mitochondrial interaction or overall intra-amoeba survival at the timepoint tested but may play a role in other aspects of *C. jejuni* persistence within Acanthamoeba. In contrast, the 11168HΔ*ciaI* mutant exhibited more than a 2-fold reduction in survival (Fig. 2a). The complemented strain (Δ*ciaI*+*ciaI*) consistently showed intermediate survival, higher than 11168HΔ*ciaI* but was not fully restored to wild-type levels, indicating a partial rescue of the intra-amoebal survival defect and suggesting that additional *C. jejuni* factors cooperate with CiaI to support full survival. Nevertheless, the 11168HΔ*ciaI* mutant failed to associate with *A. castellanii* mitochondria relative to the wild-type and complemented strains, as revealed by confocal imaging and mitochondrial aggregation size quantification (Fig. 2b–f).

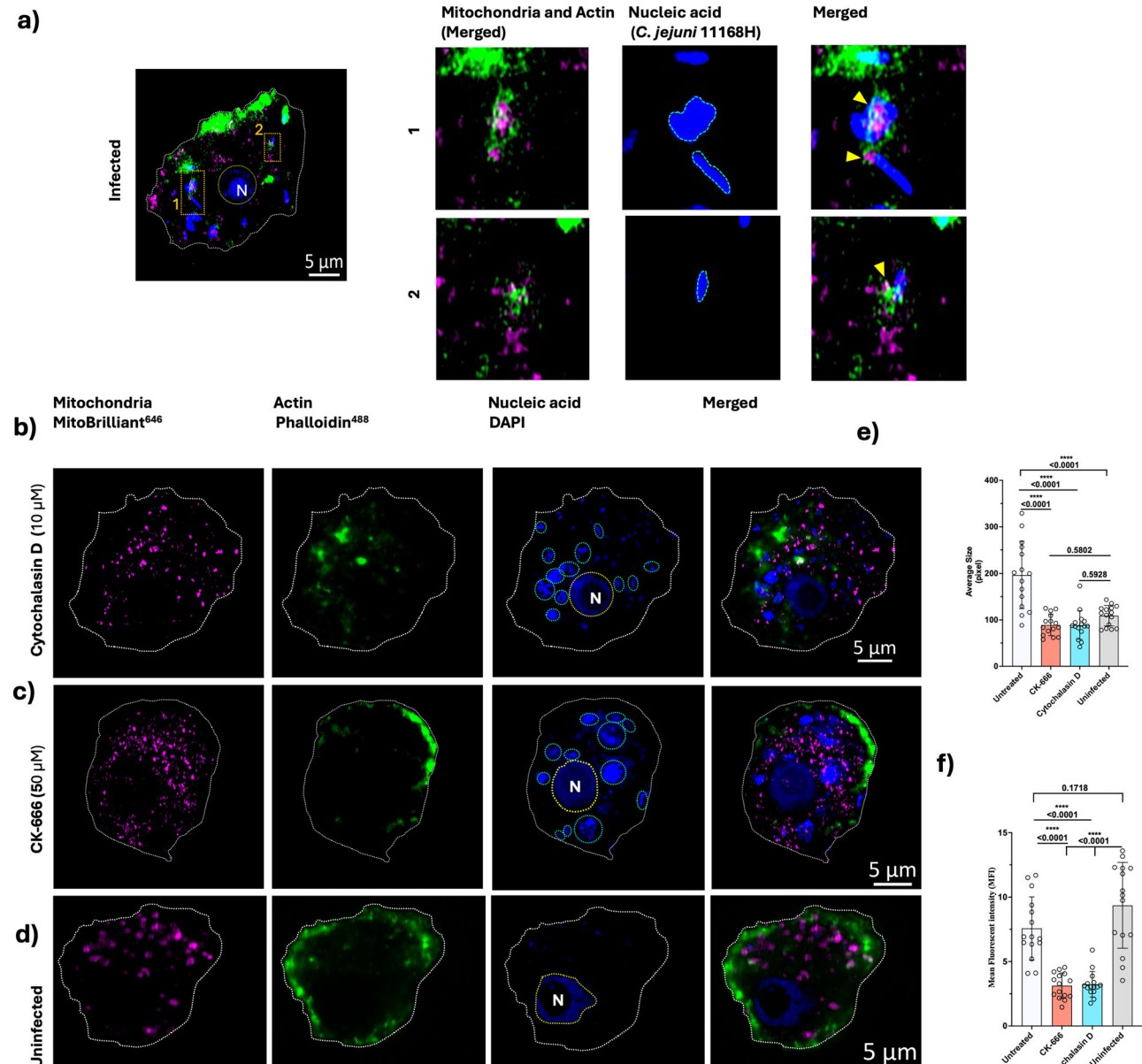

**Fig. 1 | *C. jejuni* requires host actin polymerization for mitochondrial interaction. a** Confocal microscopy of *A. castellanii* infected with *C. jejuni*. Insets highlight actin-rich regions (1, 2) where *C. jejuni* interact with mitochondria. **b** Treatment with cytochalasin D (10 μM) disrupts actin filaments and reduces bacterial-mitochondrial interaction. **c** Arp2/3 inhibition via CK-666 (50 μM) similarly diminishes actin polymerization and mitochondrial association. **d** Uninfected control cells with intact mitochondria and actin. Images acquired 4 h post-infection (including 1 h gentamicin treatment). **e** Mitochondrial aggregation was quantified as the mean mitochondrial object size (area, in pixels) per image. **f** Normalized mean

fluorescence intensity of Phalloidin[488] following Ck-666 and cytochalasin D treatment of *A. castellanii* infected with *C. jejuni* relative to untreated and uninfected. Note: In *A. castellanii*, actin commonly appears as a cortical mesh rather than thick stress fibers. This is consistent with rapid filament turnover by actin-binding proteins such as Actophorin (ADF/cofilin homologue)[31]. DAPI (Bacteria = turquoise outline; Nucleus=yellow outline); mitochondria (Violet=Mitobrilliant™ 646) and actin filaments (green = Phalloidin™ 488), yellow arrows indicate strong presence of actin, mitochondria, and bacteria in close proximity; scale bars = 5 μM (100x objective).

To determine whether CiaI alone modulates mitochondria, recombinant CiaI$_{his_6}$ (Supplementary Fig. 3a) was adsorbed onto latex beads and ~2 μg protein was incubated with *A. castellanii* for 3 h). Cells exposed to CiaI$_{his6}$-adsorbed beads exhibited fused mitochondria, while control cells treated with latex beads alone maintained punctate mitochondria similar to uninfected cells (Fig. 2g). Notably, a significant portion of the mitochondria were fused in close proximity to CiaI$_{his6}$-adsorbed beads, suggesting a localized effect. Additionally, CiaI$_{his6}$-adsorbed beads induced a substantial reduction in actin (Fig. 2h), which is critical for mitochondrial dynamics and the actin-dependent nature of *C. jejuni*-mitochondrial interactions. These findings suggest that CiaI

contributes to actin cytoskeleton rearrangement during *C. jejuni* infection. Given that CiaI is predicted to bind ATP/GTP[47,48], it may modulate host small GTPase activity to initiate or enhance actin remodelling in infected cells.

### CiaI binds nucleotide cooperatively and may function as a molecular switch

Building on the observations that CiaI influences actin remodelling and mitochondrial dynamics and given its predicted nucleotide-binding capacity, we performed structural predictions and biochemical analyses to characterize CiaI and its nucleotide interaction ability. Notably, CiaI

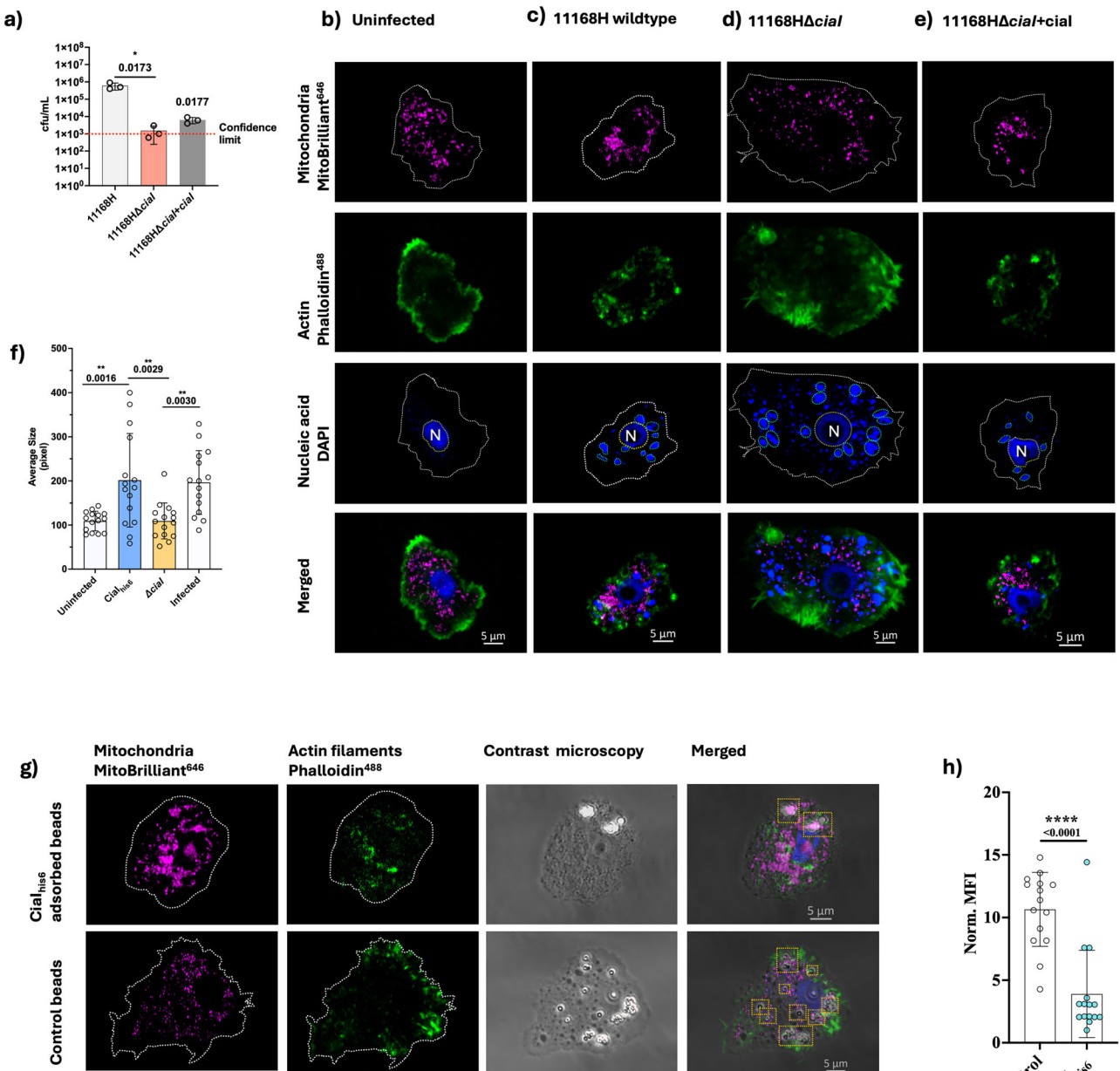

**Fig. 2 | CiaI mediates C. jejuni-mitochondrial interaction and mitochondrial aggregation. a** Survival of *C. jejuni* 11168HΔ*ciaI* mutant in *A. castellanii* (CFU/mL) after 4 h infection (*$p < 0.05$, unpaired *t*-test). **b** Confocal microscopy image showing uninfected *A. castellanii* displaying punctate mitochondria, **c** *A. castellanii* fused mitochondria associated with *C. jejuni* 11168H. **d** *C. jejuni* 11168HΔ*ciaI*-infected *A. castellanii* shows reduced mitochondria aggregation and **e** *A. castellanii* mitochondria aggregation when infected with *C. jejuni* 11168HΔ*ciaI* complemented strain Δ*ciaI*+*ciaI*. **f** Quantification of mitochondrial aggregation as the mean mitochondrial object size (area, in pixels) per image, analysed with Fiji ImageJ

($n = 15$; ** $p < 0.01$). **g** *A. castellanii* incubated with CiaI$_{his6}$-adsorbed latex beads (yellow squares) exhibit fused mitochondria while Control cells with latex beads alone retain a punctate phenotype. **h** Normalized mean fluorescence intensity of Phalloidin$^{488}$ following internalization of CiaI$_{his6}$-adsorbed latex beads, analysed with Fiji ImageJ ($n = 15$; ****$p < 0.0001$). Scale bars = 5 µM (100× objective). c.f.u Data = mean ± SD. Data represents at least three independent biological replicates DAPI (Bacteria = turquoise outline; Nucleus = yellow outline); mitochondria (Violet = Mitobrilliant™ 646) and actin filaments (green = Phalloidin™ 488). Orange boxes indicate latex beads; scale bars = 5 µM (100x objective).

contains a region (Thr$_{16}$, Lys$_{18}$, Ser$_{19}$, Lys$_{21}$) resembling a degenerate Walker-A (phosphate binding loop "P-loop") motif, which in canonical form follows the consensus GXXXXGKT/S and is essential for nucleotide phosphate binding[49]. While the classical glycine-rich elements are absent, the presence of lysine and serine residues at conserved positions supports the possibility that this region serves as a functional nucleotide-binding site. Rather than directly mediating mitochondrial aggregation, the predicted nucleotide-binding motif could allow CiaI to compete with host small GTPases that regulate actin dynamics and vesicular trafficking. Such interference may modulate cytoskeletal rearrangements and

inhibit lysosomal fusion, thereby promoting mitochondrial aggregation and bacterial persistence. We employed AlphaFold 3[50] to predict the structure of CiaI and visualize this motif. The model returned a predicted template modelling (pTM) score of 0.68, indicating moderate confidence in domain organization. Notably, the degenerate Walker A-like motif is positioned within a predicted loop region, which may provide the flexibility necessary for nucleotide binding (Fig. 3a). A sequence search using BLASTP revealed that CiaI belongs to a unique protein family, with CiaI (NF041456) identified as the only member of the superfamily cl49442[51], offering no close structural homologues.

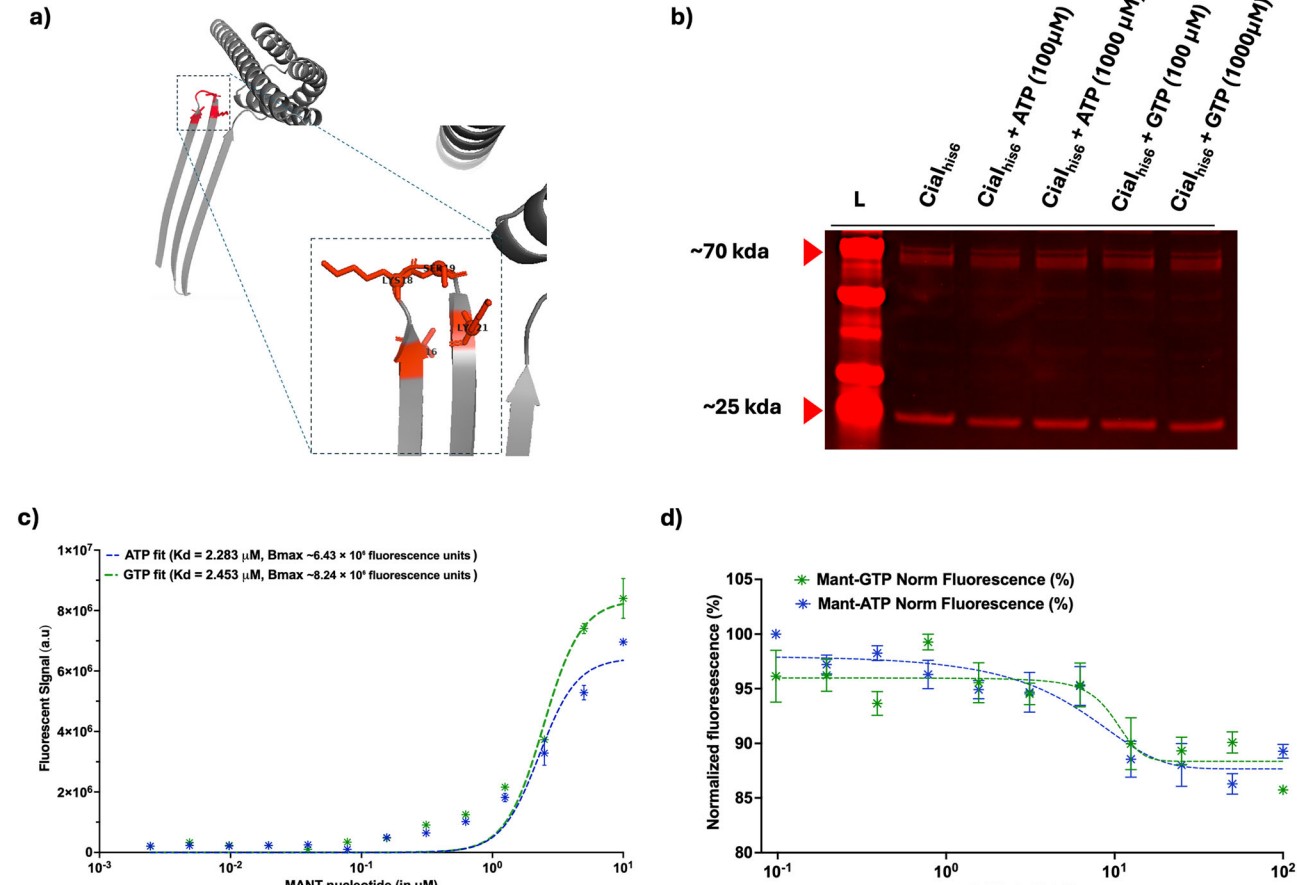

**Fig. 3 | *Nucleotide Binding Properties of CiaI Support a Co-operative Molecular Switch Model.* a** The molecular model of CiaI was predicted using AlphaFold 3 and analysed in PyMOL. Key residues Thr16, Lys18, Ser19, and Lys21 forming the degenerate Walker A-like motif within a predicted loop (inset) are highlighted in red sticks and spheres and labelled with their three-letter codes and residue numbers. The rest of the protein is shown as a grey. **b** Western blot analysis of purified CiaI$_{his6}$ (0.3 mg/mL) under denaturing conditions, showing a ~22 kDa and ~66 kDa protein bands. Samples were incubated with increasing concentrations (100–1000 µM) of ATP, GTP, or control (buffer alone); blots were probed with mouse anti-His primary antibody and detected using IRDye® 680RD goat anti-mouse IgG secondary antibody. **c** Fluorescence-based mant-nucleotide (1–10 µM) binding assay of CiaI$_{his6}$ (1 µM), modelled using a Hill coefficient of 3 to reflect cooperative binding. The calculated dissociation constants (Kd) were 2.283 µM for ATP and 2.453 µM for GTP, indicating affinity with nucleotide-dependent cooperativity. **d** Competition displacement assays using mant-labelled nucleotides and increasing concentrations of unlabelled competitors. GTP efficiently displaced mant-ATP (IC$_{50}$ ≈ 25.6 µM), while ATP weakly displaced mant-GTP (IC$_{50}$ ≈ 9 mM), indicating stronger GTP binding. Data represent at least three independent biological replicates.

Consistent with this, Western blot analysis using an anti-His antibody consistently detected the presence of both a ~22 kDa and a ~66 kDa species consistent with a monomer and trimer, respectively, regardless of nucleotide presence (Fig. 3b). Faint higher-molecular-weight species (>70 kDa) were also observed. While this does not confirm stable trimerization in solution, it indicates that CiaI can adopt multimeric states, supporting the plausibility of a trimeric assembly under physiological conditions (uncropped gels can be found in Supplementary Fig. 4a and 4b).

Guided by this, we adopted the trimeric assembly as a working model to guide analysis of nucleotide-binding. Specifically, fluorescence assays using mant-ATP and -GTP revealed a sigmoidal binding curve that was poorly fitted by a classical hyperbolic (Hill coefficient = 1) model. Instead, a cooperative binding model with a Hill coefficient of 3 (Y = Bmax * (X$^3$)/(Kd$^3$ + X$^3$)) provided the best fit to the experimental data (Fig. 3c), supporting a hypothesis that CiaI has three interacting binding sites. To assess nucleotide preference, we performed competition assays using mant-labelled nucleotides in the presence of increasing concentrations of unlabelled (cold) ATP or GTP (Fig. 3d). Cold GTP efficiently displaced mant-ATP (IC$_{50}$ ≈ 25.6 µM; logIC$_{50}$ = 1.41), indicating strong competition. In contrast, cold ATP was ineffective in displacing mant-GTP, even at high concentrations (IC$_{50}$ ≈ 8.98 mM; logIC$_{50}$ = 9.95), suggesting weak competition and potential preferential binding to GTP.

## CiaD promotes actin polymerization and acanthopodia formation

While CiaI appears to modulate host mitochondrial dynamics and actin remodelling post-internalization, the initial steps of host cell invasion likely involve distinct effectors. To understand the broader strategy by which *C. jejuni* orchestrates cytoskeletal manipulation, we turned our attention to CiaD, which was implicated in host cell entry and actin polymerization. Although differential expression of CiaD was not observed in our previous RNA-Seq dataset[44], nor detected in the proteomic profile of this study, this is consistent with its biological function as a contact-triggered effector secreted transiently during host-cell engagement. Moreover, our dual proteomic workflow enriched for host mitochondria-associated proteins, a fraction that naturally underrepresents small, soluble bacterial effectors such as CiaD, which are secreted at low abundance and lack membrane anchors. We therefore focused on CiaD due to its established role in cytoskeletal remodelling via IQGAP1 and cortactin, potent activators of the Arp2/3 complex that are stimulated in a CiaD-dependent manner[52–54].

We examined whether CiaD could influence mitochondrial morphology and found that *A. castellanii* infected with CiaD$_{his6}$-adsorbed beads exhibited normal mitochondrial morphology (Supplementary Fig. 3a; CiaD$_{his6}$ gel: Supplementary Fig. 4c). However, the cells showed a statistically significant increase in actin and enhanced acanthopodia relative to cells infected with control latex beads (Fig. 4a, b). This confirmed CiaD's role in

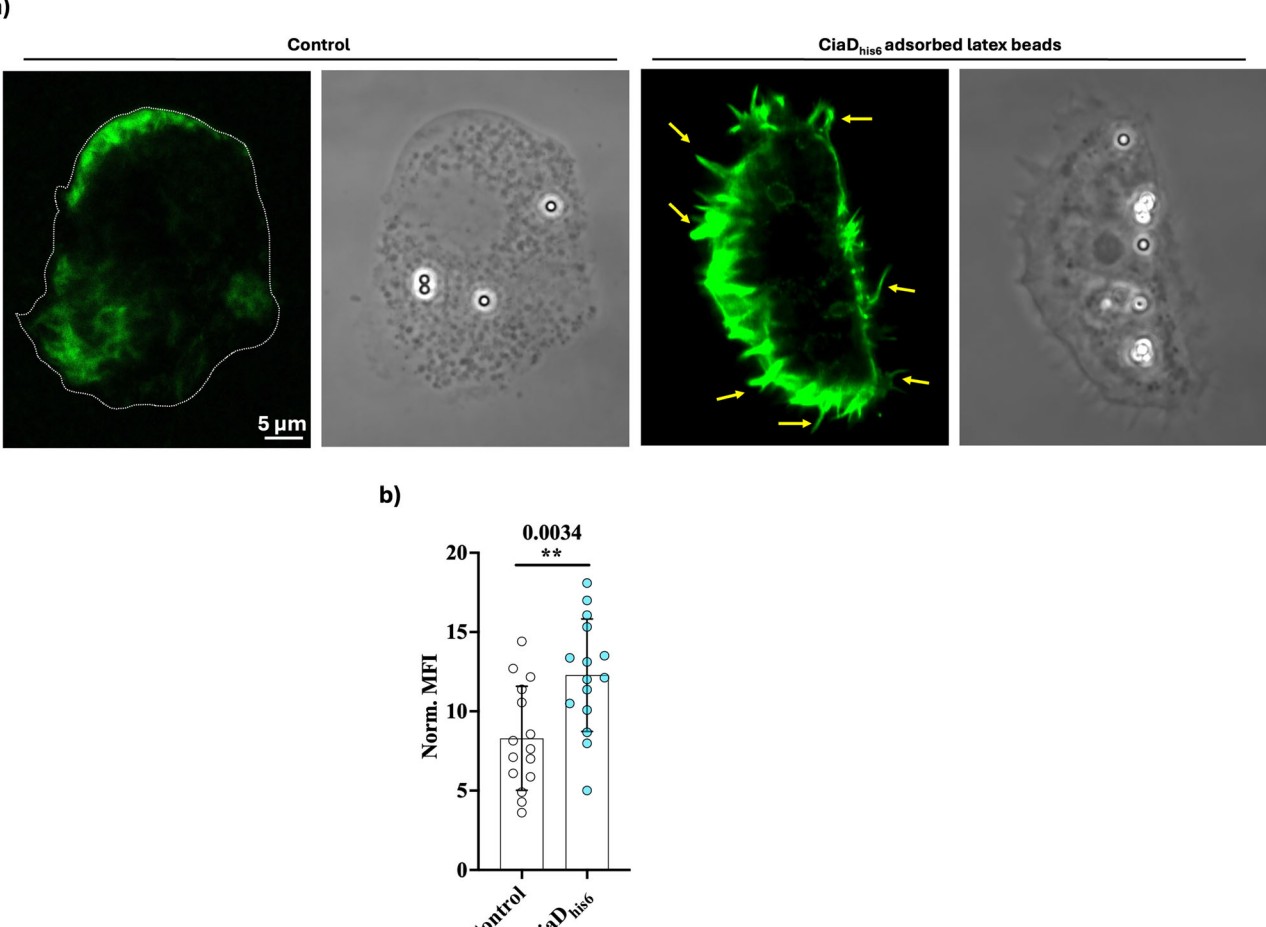

**Fig. 4 | *CiaDhis6 increases actin and enhances acanthopodia formation.***
**a** Representative images showing Phalloidin[488] treated cells highlight actin (green) and contrast microscopy images showing beads. **b** Normalized mean fluorescence intensity of Phalloidin[488] following internalization of CiaD$_{his6}$-adsorbed latex beads,

analysed with Fiji ImageJ. Scale bars = 5 μM (100x objective); Data = mean ± SD; $n = 12$; $t$-test $p < 0.05$; yellow arrows point to acanthopodia. Data represent at least three independent biological replicates.

cytoskeletal modulation. It is also worth noting that bead-adsorbed purified CiaD results in artificially high local protein concentration, producing amplified actin phenotypes not directly comparable to physiological WT infection.

### Mitochondria accumulate adjacent to actin at the *C. jejuni* entry site

Building on the demonstrated role of CiaD in modulating the host actin cytoskeleton, we next investigated how mitochondria associate with actin structures during *C. jejuni* entry, to better understand the spatial interplay between the bacterium, host mitochondria, and actin cytoskeleton in the early stages of infection.

Confocal microscopy revealed that mitochondria associate with actin structures forming the *C. jejuni* entry phagosome early in the invasion process (30 mins p.i) (Fig. 5a), consistent with CiaD-mediated cytoskeletal remodelling. To assess this across *A. castellanii* population, we quantified the spatial relationship between mitochondria (magenta) and actin (green) using Pearson's correlation coefficient (Peason's r) (Fig. 5b). Across images, Pearson's r showed modest, consistent with mitochondria and actin being related but not fully coincident during early *C. jejuni* uptake by *A. castellanii*; whole-cell analysis dilutes strong local overlap at the entry site by including non-overlapping regions. To determine whether CiaD contributes to intracellular survival or initial uptake, we compared gentamicin-protected CFU at 4 h p.i. with pre-gentamicin enumeration at 3 h. Although the Δ*ciaD* mutant displayed reduced intracellular CFU at 4 h, pre-gentamicin counts

demonstrated a significant uptake defect, indicating that the apparent "survival" phenotype arises from inefficient internalisation rather than impaired persistence. To test whether Δ*ciaD* could nevertheless be internalised when wild-type bacteria were present, we performed 1:1 co-infection assays and enumerated intracellular CFU. Despite the presence of wild-type cells, Δ*ciaD* internalisation and survival remained lower than wild type (Fig. 5c).

Consistent with this, fixed-cell imaging at 30 min p.i. showed that the Δ*ciaD* mutant interacted poorly with the amoeba surface, in contrast to the wild-type and a *ciaD* complemented, which readily formed dense surface-associated "backpacks" (Supplementary Fig. 5)[42,44]. By contrast, imaging at 4 h p.i. revealed that Δ*ciaD* bacteria that had been internalised were still capable of localising near host mitochondria (Fig. 5d), indicating that CiaD is primarily required for efficient entry rather than for subsequent mitochondrial association[52].

Proteomics analyses also revealed enrichment of iron homoeostasis-related proteins in both *C. jejuni* and *A. castellanii*, suggesting coordinated regulation of iron metabolism between host and bacteria. This interaction may have direct consequences on mitochondrial function, which is tightly linked to the host's ability to manage stress during infection.

### Iron redistribution mitigates oxidative stress and enhances *C. jejuni* survival

Building on the dynamic interplay between *C. jejuni* and mitochondria, we next investigated mitochondrial iron homoeostasis during infection.

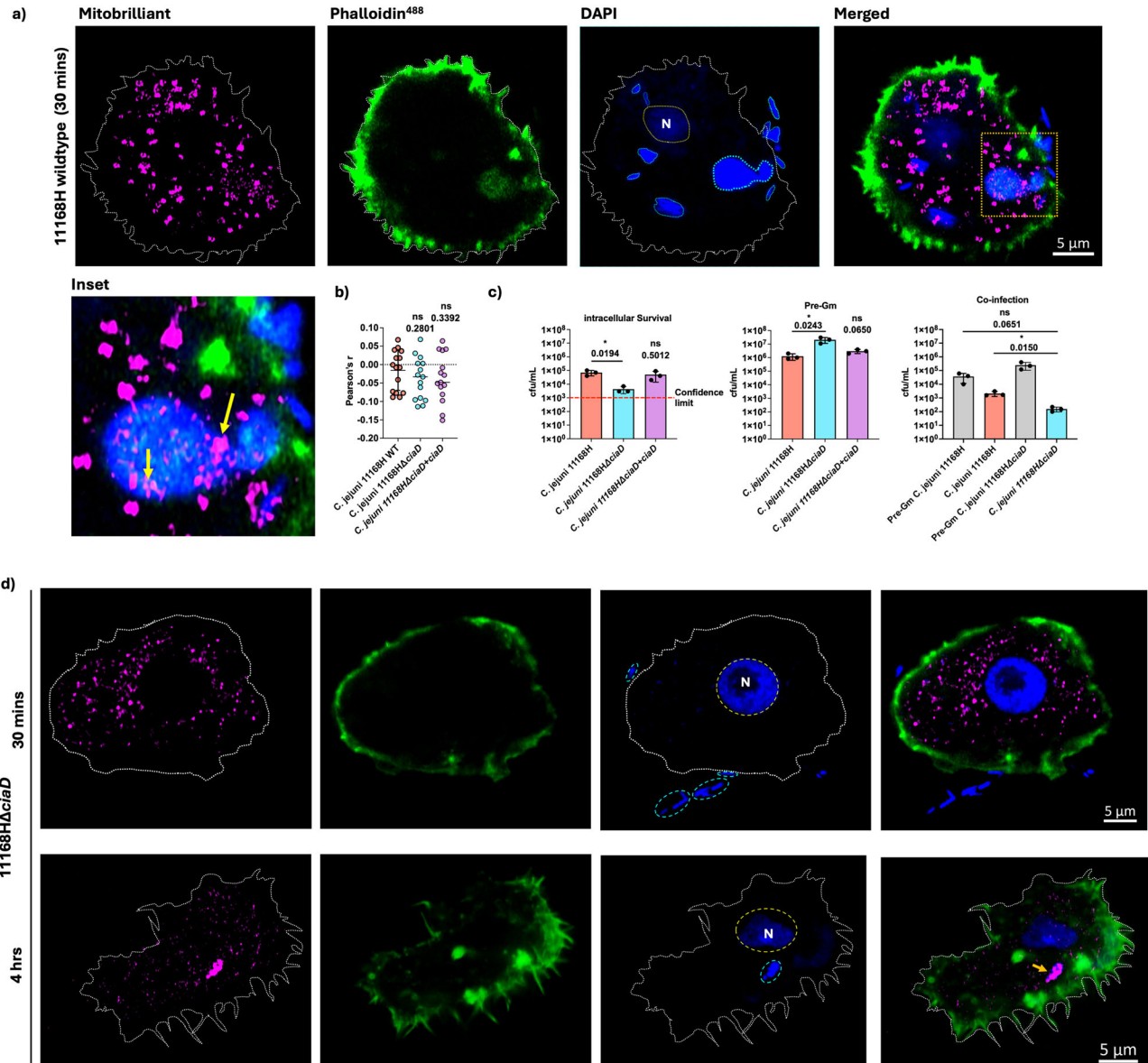

**Fig. 5 | Host mitochondria associate with actin structures during C. jejuni *invasion of A. castellanii*. a** Confocal microscopy reveals mitochondria cluster near actin-rich phagosomal structures formed during *C. jejuni* uptake (highlighted by a yellow box —inset and yellow arrows). **b** Quantification of mitochondria–actin spatial association using Pearson's correlation coefficient (Pearson's r). Gentamicin protection, pre-gentamycin, and co-infection (1:1-WT:Δ*ciaD*) assays. **d** Representative images of *A. castellanii* infected with the Δ*ciaD* mutant at 30 min and 4 h post-infection (yellow arrow shows the Δ*ciaD* mutant interacting with host mitochondria). DAPI (Bacteria = turquoise outline; Nucleus=yellow outline); mitochondria (Violet = Mitobrilliant™ 646) and actin filaments (green = Phalloidin™ 488); scale bars = 5 μM (100x objective).

High-confidence identification of multiple iron homoeostasis and oxidative stress defence proteins were detected in both organisms, prompting us to explore how modulating iron availability affects this host-pathogen interaction (Fig. 6). To assess mitochondrial iron levels, we utilized the Mito-Ferro Green assay at 4 h p.i (Fig. 6a). Flow cytometry analysis revealed a ~44% decrease in mitochondrial $Fe^{2+}$ levels in infected *A. castellanii* relative to uninfected controls, and similar trend was also observed in *A. castellanii* infected with *C. jejuni* 11168HΔ*ciaI*+*ciaI* (~34% decrease), relative to 11168HΔ*ciaI* indicating a global infection-driven restriction of the mitochondrial labile iron pool. However, confocal microscopy showed that infected cells displayed brighter Mito-Ferro Green fluorescence in mitochondria that appeared to aggregate and align with *C. jejuni* 11168H wildtype or the 11168HΔ*ciaI*+*ciaI* strain (Fig. 6b); these focal signals are indicated by the orange arrows. To capture this focal enrichment, we quantified Mito-FerroGreen across whole cells using normalized mean fluorescence intensity (MFI), which revealed higher values for cells infected with the wild-type and complemented strains compared with uninfected and Δ*ciaI* mutant-infected cells (Fig. 6c). This suggests that infection-induced mitochondrial aggregation redistributes remaining iron into localized mitochondrial clusters, creating microdomains of elevated $Fe^{2+}$ concentration, while overall mitochondria $Fe^{2+}$ remains reduced (**individual flow cytometry data are in** Supplementary Fig. 6).

Such localized iron concentration may impact oxidative stress[55], and bacterial survival[10]. Deferoxamine (DFO) was used to evaluate the impact of iron chelation versus ferric citrate (ironIII citrate) as a source of bioavailable iron supplementation on *C. jejuni* survival. Treatment with 25 μM DFO significantly ($p < 0.05$) increased *C. jejuni* survival by approximately ~25-fold compared to the untreated control, while 50 μM DFO exhibited a similar trend that did not reach statistical significance (Fig. 6c).

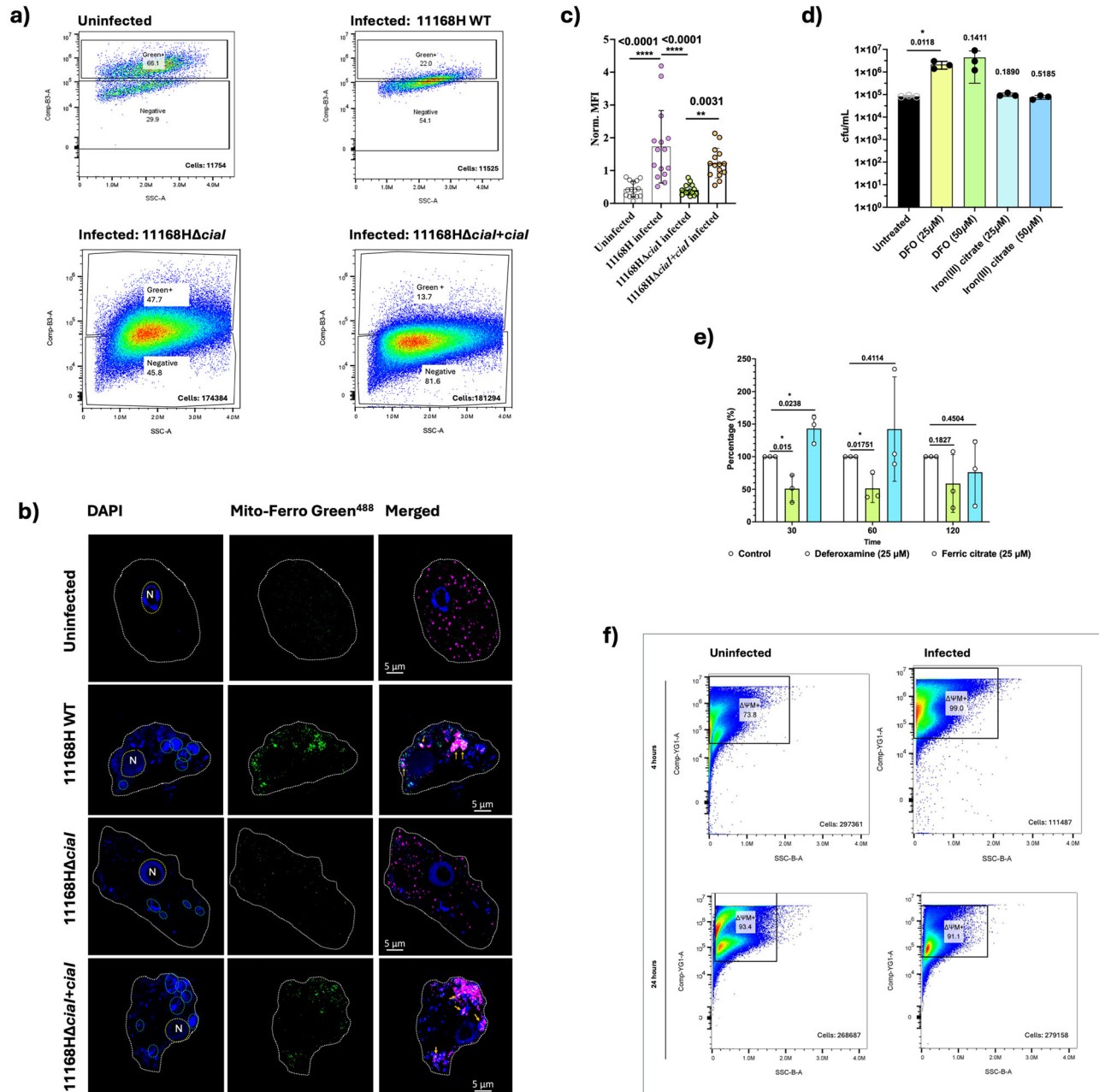

**Fig. 6 | *Iron chelation enhances C. jejuni survival by redistributing mitochondrial iron and reducing oxidative stress.* a** Flow cytometry analysis of mitochondrial $Fe^{2+}$ levels (MitoFerro Green™) in *A. castellanii* uninfected, infected with *C. jejuni* 11168H, *C. jejuni* 11168HΔ*ciaI* and 11168HΔ*ciaI+ciaI*. **b** Confocal microscopy of infected cells reveals fused mitochondria (red, Mitobrilliant™ 646) with localized iron redistribution (green, MitoFerro Green™). **c** Normalized mean fluorescence intensity of Mito-ferro Green[488] following infection, analysed with Fiji ImageJ. **d** Intracellular *C. jejuni* survival increases under iron-chelated (DFO) conditions ($p < 0.05$ vs. control). **e** ROS levels decrease with DFO treatment and transiently rise with iron

supplementation (ferric citrate). **f** Flow cytometry analysis showing TMRE fluorescence (ΔΨM) in uninfected and infected cells at 4 h and 24 h p.i. Data = mean ± SD. Data = mean ± SD; $n = 15$; $t$-test $p < 0.05$; Orange arrows indicate focal, Mito-FerroGreen enrichment in mitochondria adjacent to intracellular *C. jejuni* (WT or Δ*ciaI+ciaI*). At least 10000 cells were analysed by flow cytometry, numbers of cells are included within each panel; DAPI (Bacteria=turquoise outline; Nucleus = yellow outline); mitochondria (Violet=Mitobrilliant™ 646) and $Fe^{2+}$ (green = Mitoferro Green™ 488); scale bars = 5 µM (100x objective). All flow cytometry data are available in Supplementary Fig. 6 and Fig. 7.

We further investigated the role of iron in host oxidative burst by measuring ROS levels in *A. castellanii* under both iron-chelated and iron-supplemented conditions at 30, 60 and 120 mins relative to untreated control (Fig. 6d). DFO treatment resulted in a reduction of ROS levels by approximately 50% at both 30 minutes and 1 hr post-treatment, with a sustained (albeit slightly attenuated) reduction of ~30% at 2 h. Conversely, ironIII citrate supplementation initially increased ROS levels by ~50% at both 30 and 60 mins, although a decline of ~20% was observed by the 2 h mark.

To assess mitochondrial membrane potential (ΔΨM), we used Tetra-methylrhodamine, Ethyl Ester (TMRE), a fluorescent dye widely employed for this purpose. TMRE staining was performed on *A. castellanii* at 4 and 24 hrs post-infection (p.i.) (Fig. 6e). At 4 h p.i., infected cells exhibited a higher proportion of TMRE-high populations compared to uninfected controls, indicating an increase in ΔΨM. At 24 h p.i., TMRE fluorescence in infected cells was only 1.3% lower than in uninfected controls, and interestingly, uninfected cells at 24 h p.i. exhibited a slight increase in TMRE

fluorescence compared to 4 h (**individual flow cytometry data are in Supplementary Fig. 7**).

## Discussion

In this study, we examined how *Campylobacter jejuni* interacts with the mitochondria of *A. castellanii* to better understand how the bacterium adapts to and exploits the intracellular amoeba environment. We used a multifaceted approach that integrates dual proteomics, fluorescence microscopy, survival assays, and biochemical analyses.

While our proteomic approach was intentionally focused on mitochondria-enriched fractions, this inherently constrained our detection to proteins localized to or interacting with mitochondria during *C. jejuni* infection. Notably, mitochondrial re-localization toward the cortex during bacterial entry would be expected to increase co-isolation of myosin motors with mitochondria-enriched material. Thus, myosin signals in our dataset likely reflect both true mitochondria-proximal events and spatial enrichment caused by infection-induced organelle repositioning. Nonetheless, this spatially targeted strategy provided critical insight into the immediate molecular interface between the pathogen and host organelles. The consistent presence of actin and other actin-regulatory proteins within the mitochondrial fraction strongly suggests a functionally relevant interaction between the cytoskeleton and mitochondria. These findings informed our downstream focus on cytoskeletal dynamics in the context of mitochondrial engagement. Our proteomic data also revealed significant alterations in both host and bacterial protein profiles, particularly related to cytoskeletal remodelling, mitochondrial function, and the bacterial response to iron availability and oxidative stress. Notably, the most abundant *C. jejuni* protein identified in the mitochondrial fraction was PorA, a major outer membrane protein involved in nutrient uptake and host membrane interaction[36]. We attempted to generate a *porA* mutant to investigate its role in host mitochondrial association; however, despite multiple approaches, the mutation could not be achieved, consistent with previous reports suggesting *porA* is essential for viability[56,57]. We propose that PorA facilitates access a localized pool of charged nutrients within the mitochondrial niche, thereby enhancing bacterial persistence and its capacity to modulate host cell metabolism.

Our microscopy data show that *C. jejuni* actively exploits host actin dynamics to direct mitochondria toward the site of bacterial entry. While host-regulated processes such as mitochondrial ATP production, calcium flux, and ROS signalling can influence organelle localization[58–61], our data indicate that bacterial-driven actin polymerization plays a more immediate and decisive role in guiding mitochondria toward the forming phagosome. This is also supported by proteomic evidence revealing enrichment of actin-associated proteins during infection, underscoring the importance of cytoskeletal remodelling. We further show that branched actin networks are essential for mitochondrial association with *C. jejuni* in *A. castellanii*, suggesting that cytoskeletal rearrangements and organelle trafficking coordinate both bacterial internalization and mitochondrial engagement.

Once localized, *C. jejuni* appears to further remodel its intracellular niche, possibly by altering mitochondrial morphology and dynamics to support bacterial persistence. We show that this process is driven by Cia proteins, which have known roles in modulating host actin.

Notably, CiaD promotes host cell entry by triggering actin reorganization[52], suggesting that similar mechanisms may facilitate mitochondrial engagement.

We found that a Δ*ciaI* mutant did not interact with amoeba host mitochondria and survived significantly ($p < 0.05$) less than the wild type after 4 h infection. Interestingly, when CiaI$_{his6}$ was artificially adsorbed onto beads, led to exaggerated actin disruption and enhanced mitochondrial aggregation. In contrast to broad-spectrum actin inhibitors such as cytochalasin D or CK-666, which disrupted actin polymerization without causing mitochondrial aggregation. The observed effect suggested that CiaI$_{his6}$ likely bypassed natural regulatory mechanisms and under physiological conditions, CiaI's actin-modulating activity is spatially confined. CiaI contains a C-terminal dileucine motif (DSKKLL), which mediates

intracellular localization[62]. We propose that this motif enables CiaI to act within defined microdomains, remodelling actin at discrete sites rather than triggering widespread cytoskeletal collapse. This model is consistent with CiaI's proposed function within Campylobacter-containing vacuoles (CCVs), where it contributes to blocking vacuole maturation[62]. By modulating actin in a localized manner near the CCV, CiaI also contributes to the spatial organization of host mitochondria around *C. jejuni*, whilst preventing lysosomal fusion and supporting bacterial survival.

Based on domain architecture and biochemical evidence, we propose that CiaI functions as a nucleotide-binding protein utilizing a non-canonical Walker A motif. This supports a model in which CiaI operates as an energy-responsive molecular switch, regulated by intracellular nucleotide pools. Competitive displacement assays revealed distinct nucleotide-binding preferences and GTP sensitivity, consistent with a regulatory role responsive to cellular energy or stress[63,64]. Pull-down assays using CiaI$_{his6}$ and *A. castellanii* whole cell lysates did not reveal stable host interactors, indicating that CiaI may engage in transient or conformation-dependent interactions, rather than forming stable effector complexes. Furthermore, the lack of homology to known proteins points to CiaI as a novel effector with a unique mechanism of action. However, structural resolution will be key to determining its nucleotide-binding characteristics and interaction dynamics.

Parallel experiments confirmed CiaD's role in early actin modulation during infection. CiaD was shown to activate actin polymerization through pathways involving the Arp2/3 complex[52,53]. *A. castellanii* infected with CiaD$_{his6}$ adsorbed to latex beads showed a significant increase in actin and acanthopodia. The spiny architecture of acanthopodia enhances environmental interaction, promoting bacterial capture and phagocytosis[65]. Previously, we showed that *C. jejuni* requires functional flagella and glycosylated FlaA for interaction with *A. castellanii*[42], suggesting that initial contact is flagella-mediated. Similarly, others have reported that *C. jejuni* engages epithelial cells "flagella first", with the flagella tip contacting the host surface[20]. Fixed-cell imaging and cfu enumeration revealed that the Δ*ciaD* mutant uptake by amoeba is less efficient. However, Δ*ciaD* mutant that do become internalised can still associate with host mitochondria, indicating that CiaD is specifically required to initiate uptake rather than intracellular survival. To further test whether CiaD contributes to uptake in a bacterium-autonomous manner, we performed 1:1 co-infection assay with wild type and the Δ*ciaD* mutant. Pre-gentamicin enumeration showed that Δ*ciaD* was internalised significantly less efficiently than wild type, even when both strains were present on the same amoeba monolayer. This reduced representation of Δ*ciaD* was maintained in the intracellular CFU counts at 4 h post-infection, confirming that the defect observed in single-strain assays reflects impaired uptake rather than reduced survival. Importantly, the inability of wild-type bacteria to compensate for the Δ*ciaD* internalisation defect in mixed infections indicates that CiaD function is required in cis, acting at the level of each individual bacterium–host contact site, and cannot be complemented in trans by neighbouring wild-type cells. This interpretation is supported by our fixed-cell imaging of single-strain infections, which shows that the Δ*ciaD* mutant engaged the amoeba surface far less efficiently than the wild type at early time points (30 min p.i.). Together, these findings establish that CiaD is essential for initiating productive entry. Building on this, we propose that CiaD functions downstream of flagellar contact, promoting actin remodelling and acanthopodia extension to enhance bacterial uptake. This early and transient role also provides a mechanistic explanation for the absence of *ciaD* induction in our intra-amoebic RNA-Seq dataset.[44], as its activity precedes internalization. Time-lapse imaging provides direct evidence supporting this sequence of events (**Supplementary Movie**).

Collectively, our findings support a two-phase model of Cia effector deployment: in the first phase, CiaD drives actin polymerization to promote phagocytosis and guide mitochondria toward the forming phagosome. In the second phase, CiaI facilitates localized actin remodelling, modulating CCV dynamics[62] and promoting mitochondrial aggregation. This temporally coordinated deployment allows *C. jejuni* to couple internalization with post-entry modulation of the intracellular environment.

Our proteomic data analyses also indicated a coordinated regulation of iron homoeostasis in both the host and bacteria, an indication of a direct consequence of this interaction. Mitochondria require a constant supply of iron for essential processes such as haem biosynthesis and iron–sulfur cluster formation[66]. However, excess iron can promote the generation of reactive oxygen species (ROS) via the Fenton reaction, heightening oxidative stress within host cells[67]. Interestingly, in the presence of varying concentrations of ironIII citrate, *C. jejuni* survival was not significantly ($p > 0.05$) affected relative to the untreated control. This finding suggested that the hosts and bacteria's iron regulation mechanisms buffer additional iron, preventing significant environmental changes that could impact both *C. jejuni* and *A. castellanii* survival. A similar principle applies to DFO, where 25 μM is sufficient to chelate free iron, and increasing the concentration does not further enhance bacterial survival. Essentially, both systems may be operating near their effective limits within the tested concentration range. Nevertheless, it is evident that imposing iron limitation via chelation promotes bacterial persistence during infection, likely by mitigating ROS.

The role of iron in the host oxidative burst was assessed by measuring intracellular ROS levels, revealing that iron availability is a key determinant: chelation with DFO suppressed ROS accumulation, while supplementation with ferric citrate enhanced it. Although this assay did not directly evaluate mitochondrial function, mitochondria are central to cellular redox regulation[68] and ROS production[69,70]. It is likely that changes in iron availability impacts mitochondrial responses during infection. We therefore used TMRE staining to assess mitochondrial membrane potential ($\Delta\Psi M$). At 4 h.p.i, cells exhibited increased $\Delta\Psi M$ compared to uninfected controls. This reflects a compensatory response to the metabolic demands of *C. jejuni* infection, potentially enhancing mitochondrial function to support ATP production and counteract oxidative stress. Enhanced mitochondrial $\Delta\Psi M$ and the observed iron redistribution likely enhance cytochrome c oxidase activity, in line with our proteomic findings, to optimize electron transport while reducing electron leakage and ROS-induced damage. By 24 hours post-infection, given the total number of cells analysed, this subtle difference is more indicative of homoeostatic adaptation than mitochondrial dysfunction. A similar pattern was observed in uninfected cells, which showed a slight increase in TMRE fluorescence at 24 hours compared to the 4 hour timepoint, potentially reflecting an adaptive response to prolonged culture conditions, such as enhanced mitochondrial efficiency or a metabolic shift favouring mitochondrial stabilization.

These findings align with earlier observations of reduced ROS levels at later stages of infection[27], consistent with an adaptive host response to limit oxidative damage. Supporting this, our proteomic data also revealed high-confidence detection of mitochondrial ATP synthase subunits (ATP1 and ATP9) and host iron transporters such as Ccc1/VIT1. These changes suggest coordinated iron redistribution to sustain mitochondrial activity under infection-induced stress. Concurrently, *C. jejuni* upregulates its own iron-binding proteins, including Cft and Dps/Bfr, as well as oxidative stress defence proteins KatA and Tpx. This reflects a bacterial strategy to balance iron acquisition, suppress oxidative damage, and ensure persistence within the host cell.

## Perspective

Building on our experimental findings, we propose a model in which *C. jejuni* manipulates host actin dynamics for dual purposes: to trigger phagocytosis and to promote mitochondrial aggregation and iron homoeostasis, thereby enhancing intracellular persistence (Fig. 7). This two-step modulation of actin dynamics, initiated by CiaD-induced polymerization, followed by CiaI-mediated depolymerization, links cytoskeletal remodelling to mitochondrial metabolism.

We suggest that *C. jejuni* fine-tunes host cell conditions by sequentially engaging these pathways, offering new insights into how bacterial pathogens adapt their intracellular niche. While recent studies have highlighted pathogen-driven organelle manipulation[60,71], our model

introduces the novel concept of sequential mitochondrial remodelling—mediated by actin—and coupled to iron regulation as a survival strategy. Our earlier observation that *C. jejuni* remains associated with mitochondria during encystment[27], further supports the idea of active subversion of host organelle function rather than incidental proximity. This sustained interaction may facilitate amoeba cyst formation and long-term persistence within this protozoan host. Indeed, mitochondrial structural changes have been identified as early features of encystment in Acanthamoeba[72].

## Conclusion

A key limitation of our study was the inability to assess canonical regulators of mitochondrial aggregation, such as Drp1 phosphorylation and OPA1 processing, because *A. castellanii* mitochondrial dynamin-related GTPases and fusion/fission regulator proteins exhibit substantial sequence and structural differences compared to opisthokont orthologs. For instance, *A. castellanii* Drp1 (L8HLM2) lacks conserved phosphorylation sites, and poor antibody cross-reactivity precluded standard detection approaches. There is a need for improved molecular tools and characterization in this under-studied yet medically relevant amoeba.

In addition, while DAPI staining provided a practical means to visualise intracellular *C. jejuni* in fixed cells, this approach labels all DNA-containing structures and therefore cannot unambiguously distinguish bacteria from host-derived nucleic acid. Future live-cell imaging approaches using bacterial pre-labelling or genetically encoded reporters will be valuable for resolving the temporal dynamics of actin remodelling and mitochondrial repositioning during infection. Finally, our results support the hypothesis that *C. jejuni* uses a two-step actin-mediated strategy, orchestrated by CiaD and CiaI, to manipulate mitochondrial dynamics and iron availability within *A. castellanii*. This interaction likely provides metabolic benefits—such as access to iron, nutrients, and protection from oxidative stress—supporting persistence outside warm-blooded hosts.

## Methods

### Bacteria and amoebae cultures

Bacteria were stored using Protect bacterial preservers (Technical Service Consultants) at − 80 °C. *C. jejuni* strains were streaked on blood agar (CBA) plates containing Columbia agar base (Oxoid) supplemented with 7% (v/v) horse blood (TCS Microbiology) with or without selective antibiotics, and grown at 37 °C in a microaerobic chamber (Don Whitley Scientific), containing 85% $N_2$, 10% $CO_2$, and 5% $O_2$ for 48 h. Bacteria were grown for a further 16 h at 37 °C prior to the experiment.

*Acanthamoeba castellanii* CCAP 1501/10 (Culture collection of Algae and protozoa) was cultured to confluence at 25 °C in peptone, yeast and glucose (PYG)[23]. Viability was determined using light microscopy.

### Mutant generation in *C. jejuni* 11168H

*C. jejuni* 11168HΔ*ciaC*, Δ*ciaI* and Δ*ciaD* mutants were generated using isothermal assembly (ISA) cloning as described previously[42]. The genes of interest were disrupted by inserting apramycin resistance (Apr) cassettes into the reading frame. The constructs were initially cloned in *E. coli* DH5α, selected on LB agar containing apramycin and ampicillin. Plasmids, pgem®3zf⁺, containing the upstream and downstream gene flanks were recombined into the *C. jejuni* 11168H chromosome via double homologous recombination following electroporation and antibiotic selection. Successful mutants were confirmed by PCR, with primers listed in Supplementary Table 1.

Complementation of the Δ*ciaI* and Δ*ciaD* mutants was generated using the plasmid pRRH (Hygromycin) as previously described[73]. Briefly, the wild-type *ciaI* or *ciaD* gene, along with its intact native promoter or chloramphenicol promoter, respectively, was amplified and the fragment was inserted into the pRRH plasmid. The construct was then electroporated into *C. jejuni* 11168HΔ*ciaI* and 11168HΔ*ciaD* to make 11168HΔ*ciaI*+*ciaI* and 11168HΔ*ciaD*+*ciaD*, respectively.

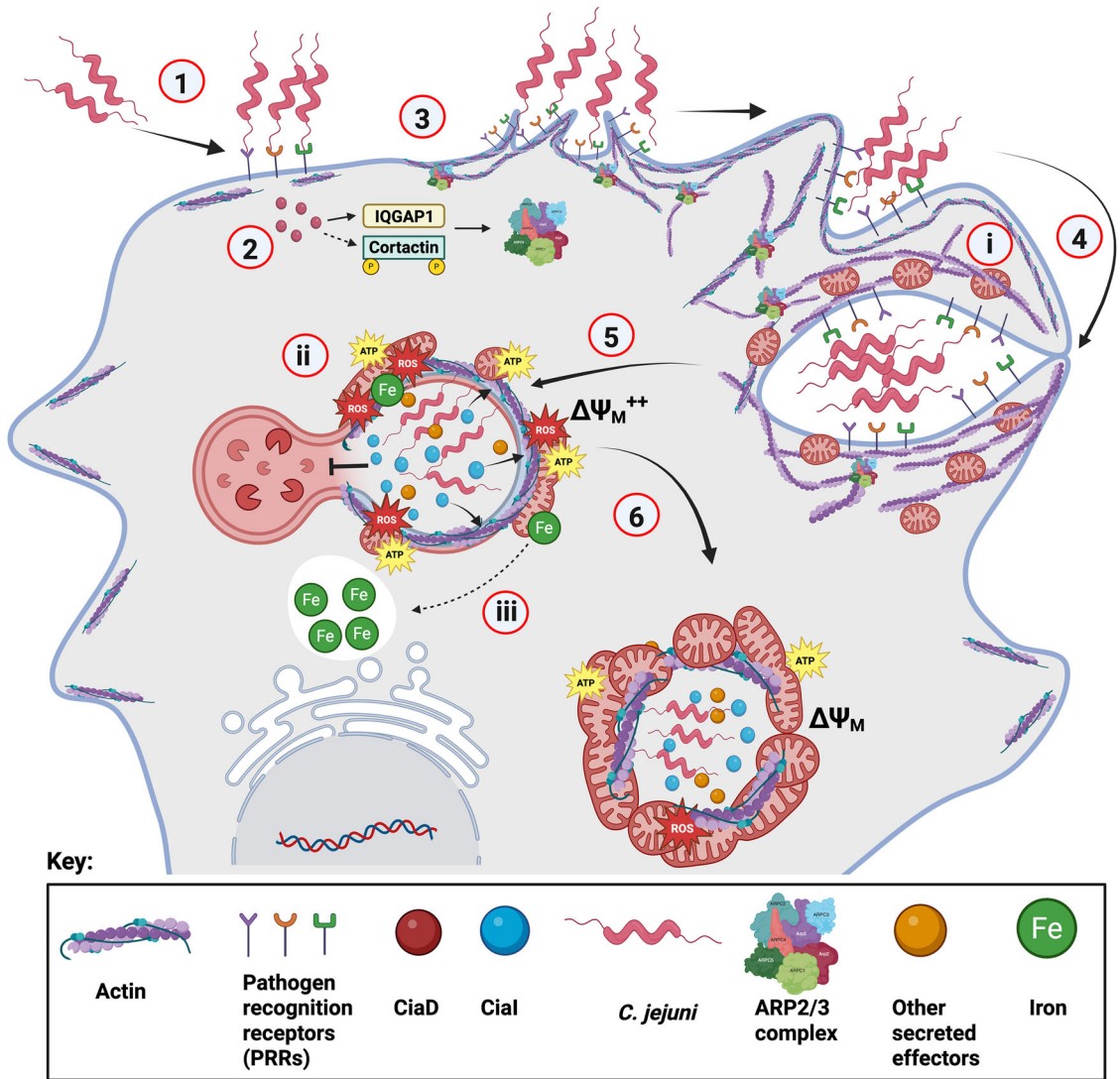

**Fig. 7 | *Hypothetical Model of Actin Modulation by C. jejuni to promote mitochondrial aggregation and iron homoeostasis, for intracellular survival and persistence.* 1**. Attachment to *A. castellanii*: *C. jejuni* approaches *A. castellanii* and binds to yet-unknown pathogen recognition receptors (PRRs)[74], utilizing its glycosylated flagella for attachment[42]. **2**. Secretion of CiaD Protein: Upon contact, *C. jejuni* secretes the CiaD effector protein, initiating host signalling cascades[52]. **3**. CiaD-mediated Actin Polymerization: CiaD promotes actin polymerization via IQGAP1 and cortactin-dependent Arp2/3 pathway[53,54]. This cytoskeletal rearrangement facilitates acanthopodia formation and enhances the surface area, aiding in phagocytic cup development. **4**. Internalization of *C. jejuni*: During this process, (i) the resulting phagosome associates with host mitochondria in an actin-dependent manner. **5**. Transition to Campylobacter-Containing Vacuole (CCV) and Actin Depolymerization: As the phagosome matures into a CCV, (ii) *C. jejuni* secretes

CiaI, which inhibits phagosome-lysosome fusion[62] by promoting localized actin depolymerization. This subsequently remodels mitochondrial dynamics, leading to aggregation and iron redistribution. The localized accumulation of iron near the CCV triggers ROS production via Fenton chemistry. Simultaneously, mitochondrial membrane potential ($\Delta \Psi M$) and ATP levels rise, *A. castellanii*, (iii) exports iron from mitochondria into vacuolar compartments via the Ccc1/VIT1 transporter system[35], concurrently, *C. jejuni* upregulates iron sequesters and oxidative stress defences to buffer increased ROS. **6**. Enhanced *C. jejuni* persistence: This creates a long-term favourable niche for *C. jejuni* survival—even following *A. castellanii* encystation[27]. Dashed arrows indicate hypothesized pathways. $\Delta \Psi M++$ = increased membrane potential; $\Delta \Psi M$ = normal membrane potential (Fig. generated using Biorender (biorender.com)).

## CiaI and CiaD expression of recombinant protein

Recombinant *C. jejuni* CiaI$_{his6}$ and CiaD$_{his6}$ protein expression and purification were performed as follows: ciaI and *ciaD* genes were cloned into the pET21a $^{(+)}$ plasmid using a NEBuilder HiFi DNA assembly cloning kit. The recombinant proteins were overexpressed in *E. coli* strain BL21(DE3) cultured in LB broth containing 150 µg/ml ampicillin until ~$OD_{600}$ = 0.6, isopropyl β-D-1-thiogalactopyranoside (IPTG) at 1 mM was added to the culture and incubated for a further 4 h at 37 °C. Cells were harvested, resuspended in buffer A (20 mM Tris-HCl, 500 mM NaCl and 20 mM imidazole at pH 8) with protease inhibitors, for lysis. The supernatant was incubated with Ni-NTA agarose, and the recombinant proteins were eluted using buffer B (20 mM Tris-HCl,

500 mM NaCl, 400 mM imidazole at pH 8). Primers used for cloning are listed in Supplementary Table 1. Recombinant proteins were dialysed in buffer C (20 mM Tris-HCl, 200 mM NaCl and 10% glycerol, pH 8) for 24 h.

## Recombinant protein latex beads adsorption

Dialysed recombinant CiaI$_{his6}$ and CiaD$_{his6}$ were chemically adsorbed onto ~1 µm latex beads (Sigma-Aldrich) as previously described[27]. Briefly, beads were prepared by washing with borate buffer (0.1 M, pH 8.5) three times. Up to 400 µg of protein was added to the washed buffer and incubated at room temperature overnight. Protein adsorbed beads were centrifuged, and the supernatant was retained to determine the concentration of adsorbed

protein. Latex beads were resuspended in 5% BSA in borate buffer for 30 mins, and then washed and resuspended in PBS.

### Invasion and survival assay

Invasion and survival assays were performed as previously described[24,44]. Briefly, *C. jejuni* 11168H was incubated with a monolayer of ~$10^6$ *A. castellanii* at a multiplicity of infection (m.o.i) of 100 for 3 h in PYG media at 25 °C in the presence of deferoxamine (DFO) in dimethyl sulfoxide (DMSO) or ferric citrate in water (at the concentration stated in the results). The monolayer was washed three times with PBS and incubated for 1 h in PYG media containing 100 µg ml$^{-1}$ gentamicin. Bacterial cells were harvested by lysing amoebae in PBS containing 0.1% (v/v) Triton X-100 for 10 min at room temperature. The suspension was centrifuged for 10 min at 4000 *g*, and the resultant pellet was resuspended in 1 ml PBS and enumerated for colony-forming units.

The concentrations for cytochalasin D (10 µM) and the subsequent Arp2/3 inhibitor CK-666 (50 µM) were chosen based on preliminary experiments demonstrating that these doses do not inhibit *C. jejuni* invasion of *A. castellanii*.

### Co-infection assay

A 1:1 mixture of WT *C. jejuni* 11168H and the Δ*ciaD* mutant (distinguished by apramycin resistance) was prepared at a combined MOI of 100 and used to infect ~$10^6$ *A. castellanii* cells in PYG medium. Prior to gentamicin treatment, extracellular bacteria were plated on non-selective CBA and apramycin-supplemented CBA to determine the ratio of WT to Δ*ciaD*. Following 1 h gentamicin protection (100 µg ml$^{-1}$), amoebae were lysed in PBS + 0.1% Triton X-100, and lysates were plated in parallel on non-selective and apramycin-selective CBA. Intracellular WT CFU were calculated as: (CFU on non-selective agar) – (CFU on apramycin agar), while Δ*ciaD* CFU were taken as the CFU on apramycin plates.

### Liquid chromatography Mass spectrometry

Lysates (in RIPA buffer) were aliquoted and precipitated with the cold acetone protocol. The protein pellets were resuspended in 100 mM triethylammonium bicarbonate buffer (TEAB) and tryptically digested at a ratio of 100:1 (protein: trypsin) at 37 °C for overnight, after being treated with Dithiothreitol (DTT) regent for reduction followed by iodoacetamide (IAA) alkylation. The digested peptides were desalted with a C18 spin column and dried by SpeedVac prior to LCMS analysis. The tryptic peptides were directly ionized within the Easy-spray ion source (Thermo) and injected into Orbitrap Eclipse Tribrid mass spectrometry (Thermo Fisher Scientific) coupled with Ultimate 3000 RSLC nano system for analysis. For liquid chromatography, a reverse-phase Thermo Acclaim Pepmap trap column (2 cm length, 75 µm in diameter and 3 µm C18 beads) was connected to the nanoflow HPLC on an Easy-spray C18 nano column (50 cm length, 75 µm in diameter, Thermo Fisher Scientific). Buffer A (5% ACN, 0.1% formic acid) and buffer B (80% ACN, 0.1% formic acid) were used. Peptides were eluted with a 60 min gradient, ramping from 4% to 10% to 5 min, from 10% to 30% to 37.5 min, from 30% to 40% to 40 min, from 40% to 99% to 42.5 min, kept at 99% to 46.9 min and decreasing to 4% at 47 min. The MS instrument was operated in the positive ion mode with an electrospray through a heated ion transfer tube at 275 °C. MS DIA datasets were acquired within Xcalibur 4.7 using the following parameters: scan range 400-900 m/z, MS resolution of 60,000 at m/z 200, a normalized AGC target (%) of 250, and maximum injection time of Auto. The MS/MS scan was performed in HCD mode with the following parameters: using a 12 Da isolation window with a 1 Da overlap over 400–900 *m/z* precursor and a scan range of 145–1450 *m/z*, Orbitrap resolution 15,000 with maximum injection time of Auto, a normalized AGC target (%) 800, and normalized collision energy = 30%. All data were acquired in positive polarity and centroid mode.

Resulting DIA raw files were searched against *A. castellanii* (strain ATCC 30010 / Neff) or *C. jejuni subsp. jejuni* O:2 (strain ATCC 700819/ NCTC 11168), following the analysis pipeline within PEAKS Studio

software (Bioinformatics Solutions Inc, version 12). DIA DB search parameters: precursor and fragment mass error tolerances (auto-detected) with match between run, trypsin as enzyme with 2 miss cleavage, cysteine carbamidomethylating as a fixed modification and oxidation on methionine as a dynamic modification, report filter as precursor -10lgP >= 20. Label-free quantification was applied here with DIA LFQ workflow embedded, using high precision mode, default filter setting for peptide and protein and TIC normalization.

### Imaging

Confocal laser scanning micrographs were obtained using an inverted Zeiss LSM 880 confocal microscope (Zeiss).

Adherent *A. castellanii* (~$10^6$) in PYG media were infected with *C. jejuni* 11168H at an m.o.i of 100 in 35 mm µ-Dish devices (IBIDI). After 3 hours of infection, cells were washed with warm PBS and incubated for 1 h in PYG media containing 100 µg ml$^{-1}$ gentamicin and MitoBrilliant[646] (excitation and emission wavelengths of 648/662 nm). Infected *A. castellanii* cells were fixed with 4% paraformaldehyde (PFA) for 20 mins, cells were washed three times and permeabilised with 0.1% Triton X-100 for 5 mins. Cells were washed with PBS and incubated with Phalloidin[488] (Thermofischer), excitation and emission wavelengths of 488/510 nm.

Actin disruption was carried out as above, with some differences as follows. *A. castellanii* were treated with cytochalasin D (Sigma-Aldrich) and CK-666 (Sigma-Aldrich) 30 mins before infection, and treatment was maintained throughout the experiment.

### Fluorescent nucleotide binding assay using Mant-labeled nucleotides

Binding of nucleotides to recombinant CiaI$_{his_6}$ was assessed using fluorescent N-methylanthraniloyl (mant)-labeled nucleotides. Reactions were performed in black, flat-bottom 96-well plates (Thermofischer) in a final volume of 100 µL per well. Unless otherwise stated, each well contained 1 µM purified CiaI$_{his_6}$ in binding buffer (50 mM Tris-HCl pH 7.5, 150 mM NaCl, 5 mM MgCl$_2$). Increasing concentrations (0.1–20 µM) of mant-ATP or mant-GTP (Jena Bioscience) were added, and the mixture was incubated at room temperature for 20 minutes in the dark to allow equilibrium binding.

Fluorescence was measured using a SpectraMax iD5 plate reader (Molecular Devices) with excitation at 355 nm and emission at 448 nm. Background fluorescence from mant-nucleotide alone (in buffer without protein) was subtracted from each reading. Normalized fluorescence values were plotted against ligand concentration, and binding curves were fitted using a nonlinear regression model with a Hill coefficient of 3 (GraphPad Prism 10) to estimate apparent dissociation constants (Kd). Each condition was repeated in at least three independent experiments.

### Competition displacement assay

To assess nucleotide-binding specificity, we performed competition assays in which mant-labelled nucleotide binding was challenged with increasing concentrations of unlabelled (cold) ATP or GTP. Reactions were prepared in 96-well plates with 1 µM CiaI$_{his_6}$ pre-incubated with 10 µM mant-ATP or mant-GTP in binding buffer (50 mM Tris-HCl, pH 7.5, 150 mM NaCl, 5 mM MgCl$_2$). After a 20 min incubation at room temperature, increasing concentrations of unlabelled ATP or GTP (ranging from 1 µM to 100 µM) were added, followed by a further 20 min incubation in the dark. Fluorescence was recorded as described above. Fluorescence signals were normalized to the no-competitor control (set to 100%), and displacement curves were generated using a four-parameter logistic model to determine IC$_{50}$ values. Hill slopes were also calculated to assess binding cooperativity. Each condition was repeated in at least three independent experiments.

### Western blot

CiaI$_{his_6}$ (0.3 µg/µL) were incubated with increasing concentrations (100–1000 µM) of ATP, GTP, or control (buffer alone) prior to SDS-PAGE. Following SDS-PAGE electrophoresis, proteins were transferred to a

nitrocellulose membrane (Amersham™ Protran® Western blotting membranes) and probed with mouse anti-His primary antibody (1:2000 dilution), followed by IRDye® 680RD-conjugated goat anti-mouse IgG secondary antibody (1:10,000 dilution). Blots were imaged using Licor Odyssey M (LI-COR).

## MitoFerro green and Tetramethylrhodamine, ethyl ester (TMRE) assay

For mitochondrial ferrous iron detection, MitoFerro Green (Dojindo) was used. A 1 mM stock solution was prepared according to the manufacturer's instructions (dissolved in ethanol). After the 3-hour infection step and the subsequent 1-hour gentamicin treatment, *Acanthamoeba castellanii* were washed with warm PBS. Cells were then incubated with 5 μM MitoFerro Green in PBS for 30 minutes at 25 °C in the dark. Following incubation, cells were washed three times with PBS, fixed with 4% paraformaldehyde (PFA) for 20 minutes, and permeabilized with 0.1% Triton X-100 in PBS for 5 minutes. After washing three times with PBS, cells were prepared for imaging (MitoFerro Green excitation and emission wavelengths of 505 nm/535 nm).

For mitochondrial membrane potential assessment, Tetramethylrhodamine, ethyl ester (TMRE) was used. After the same infection and gentamicin treatment, cells were incubated with 50 nM TMRE in PBS for 30 minutes at 25 °C in the dark. Following incubation, cells were washed three times with PBS.

For flow cytometry analysis of both assays, stained cells were resuspended in 200 μL of PBS and analysed using the Cytek Aurora spectral flow cytometer. The instrument was configured with the following settings: 488 nm (blue) laser excitation for both MitoFerro Green and TMRE, with emission detection using a 525/40 nm filter for MitoFerro Green and a 575/30 nm filter for TMRE. Voltage settings were optimized based on unstained controls. Forward scatter (FSC) and side scatter (SSC) parameters were used to exclude debris and distinguish *A. castellanii* populations. MitoFerro Green-positive cells were identified based on fluorescence intensity in the FITC channel, and TMRE-positive cells were identified in the PE channel. A minimum of 10,000 events per sample were acquired, and data were analysed using FlowJo software. The percentage of MitoFerro Green- and TMRE-positive cells, along with the mean fluorescence intensity (MFI) for both, were quantified. Data available in Supplementary Fig. 7.

## Image analysis (Fiji/ImageJ)

Full, uncropped fields of view were analysed; actin was green, mitochondria red/magenta, bacteria blue. All processing was done in Fiji (ImageJ) with identical settings. RGB images were split into single channels: red (mitochondria) and green (actin). When multiple cells were present, whole-cell ROIs were drawn and applied consistently; otherwise, the entire field was analysed.

## Co-localization

Pixel-wise colocalization between mitochondria and actin was quantified as Pearson's correlation coefficient (Pearson's r) using JACoP with default (no-threshold) settings, yielding one value per image/ROI.

## Actin mean fluorescence intensity (MFI)

Actin intensity was quantified on the green channel after the same background subtraction. For each image/ROI, the mean gray value was obtained via Analyse Measure. A local background ROI (cell-free area) was measured in the same field and subtracted from the actin mean to give background-corrected MFI. Exposure/gain and processing were held constant; images with saturation or artifacts were excluded.

## Mitochondria

Mitochondrial aggregation was quantified from the red channel. For each image/ROI, the mitochondrial channel was exported from Fiji as a grayscale image and analysed using UTHSCSA ImageTool for Windows v3.0 (University of Texas Health Science Center, San Antonio, TX,

United States). The same whole-cell ROIs (or full-field ROIs when only one cell was present) defined in Fiji were used for all conditions. Within each ROI, mitochondrial objects were detected, and the mean object size (area, in pixels) was recorded, yielding one value per image/ROI. Because Acanthamoeba mitochondria are normally punctate, dispersed mitochondria appear as numerous small objects, whereas aggregated mitochondria form fewer, larger objects; accordingly, an increase in mean mitochondrial object size reflects increased mitochondrial aggregation.

## Statistics and reproducibility

Data are presented as mean ± standard deviation (SD). GraphPad Prism 10 was used for all statistical analyses. For two-group comparisons, unpaired t-tests were applied, while one-way ANOVA followed by appropriate post-hoc tests was used for multiple comparisons where applicable. Nonlinear regression models were used to fit fluorescence-based nucleotide binding and competition displacement data. Dissociation constants (Kd) were estimated using a Hill equation (variable slope, Hill coefficient = 3) for binding curves, while $IC_{50}$ values and Hill slopes were derived from four-parameter logistic regression fits of competition assays. Flow cytometry and confocal microscopy data were analysed for significance using appropriate parametric or non-parametric tests depending on distribution. All CFU experiments are presented as three biological replicates unless otherwise stated. All statistical tests were performed with a confidence threshold of $p < 0.05$ unless otherwise stated. Measurements were taken from biological replicates (separately infected *A. castellanii* cultures or independently purified recombinant proteins). No repeated measurements from the same sample were used unless specified for technical replicates. All numerical source data underlying the graphs and charts presented in this study are provided in Supplementary Data 4.

## Reporting summary

Further information on research design is available in the Nature Portfolio Reporting Summary linked to this article.

# Data availability

The mass spectrometry proteomics data have been deposited to the ProteomeXchange Consortium via the PRIDE partner repository with the dataset identifier PXD065850, and all analysed data underlying Table 1 can be found in Supplementary Data 1 and 2. All biological replicate flow cytometry data presented in this study can be found in the supplementary Information. Source values underlying all graphs presented in the study can be found in supplementary data 3.

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

## Acknowledgements
We thank Christian Chiu for providing training, technical assistance and analyses using flow cytometry. We thank Steven Lynham and Xiaoping Yang from the Centre of Excellence for Mass Spectrometry at King's College London for proteomic analyses. We thank Professor Serge Mostowy for his advice. We also acknowledge the Imaging and Cytometry Platform for Infection Biology (LSHTM). This work was supported by the Biotechnology and Biological Sciences Research Council Institute Strategic Programme BB/R012504/1 constituent project BBS/E/F/000PR10349 to B.W.W.

## Author contributions
F.N. conceptualized, designed, performed experiments and wrote the first draft; F.N. and B. W.W. edited the final draft of the manuscript

## Competing interests
The authors declare no competing interests.

## Additional information

**Peer review information** . *Communications Biology* thanks Huan Lian and the other anonymous reviewer(s) for their contribution to the peer review of this work. Primary Handling Editor: Tobias Goris. A peer review file is available.

