## [Transparent Peer Review file · Communications Biology]

A two-step actin-mediated strategy enables *Campylobacter jejuni* to promote mitochondrial aggregation and iron homeostasis, for intracellular survival and persistence

Corresponding Author: Dr Fauzy Nasher

Version 0:

Reviewer comments:

Reviewer #1

(Remarks to the Author)

Here, the authors present an interesting two phase model of *C. jejuni* infection of *A. castellanii*:

1. Firstly the *C. jejuni* effector CiaD drives actin polymerization to promote phagocytosis and guide mitochondria towards the forming phagosome.
2. Secondly, the *C. jejuni* effector CiaI facilitates localized actin remodelling, modulating CCV dynamics and promotes mitochondrial fusion.

The summation being that his temporally coordinated deployment allows *C. jejuni* to couple internalization with post-entry modulation of the intracellular environment. While the data provided supports this, there requires a significant amount of improvement to data analysis approaches.

My comments and concerns on data are below:

Comments and suggestions:

Results:

Table 1.

- The authors conduct LC-MS/MS on mitochondrial enriched fractions to look at proteins associated with mitochondria during infection. It is evident that *C. jejuni* infection significantly alters the mitochondrial spatial distribution and morphology within the *A. castellanii* cell. Therefore, some of the changes observed in host associated proteins may just be due to spatial localisation and not due significant changes in interactions. For example, mitochondrial relocation from perinuclear to the cellular periphery will naturally increase pulldown of cortical actin components. Could the authors please comment?
- It would be beneficial to be able to visualise these changes in particular to host protein changes in relation to the total detected proteome within the fractions. I would suggest visualisation of the data in a more modern approach such as a volcano plot for the readers benefit.
- (140) How similar are the genes of mammalian actin cytoskeleton to *A. castellanii*. Could the authors comment on how well *C. jejuni* Cia proteins would be expected to interact with both?

Figure 1.

- It is problematic to use DAPI for *C. jejuni* staining in cells in which host nuclei, nuclear bodies and host DNA will also be stained. How do the authors know what is actually *C. jejuni*? Ideally a pre-labelling approach such as a viability stain would be beneficial
- A major issue is with the interpretation of qualitative imaging data. The phalloidin actin cytoskeleton does not look indicative of filamentous cortical actin even in the untreated *A. castellanii*.
- To interpret the changes in mitochondrial distribution and actin. The authors will need to employ quantitative image analysis approaches to make statements such as 'significantly reduced' (line 192).
- Examples of quantification needed include: 1) quantification of size/intensity of mitochondria and actin puncta and intensity distribution. 2) colocalization analysis of segmented *C. jejuni* with *A. castellanii* mitochondria and comparison with CK-666 and Cytochalasin D treatment
- Does Cytochalasin D and CK-666 treatment alter *C. jejuni* cell entry? Could this be also affecting the downstream affects? Quantification of *C. jejuni* burden within the cell and correlation to effect sizes would be beneficial

Figure 2.

- The previous comments about single cell image analysis stand for figure 2.
- TO make this cleared WT *C. jejuni* should also be shown to directly compare the CiaHis6.
- As Mitobright is a live cell dye. This figure would be significantly helped if the author tracked mitochondrial fusion at the sites of the CiaHis6 beads over time and compared to latex bead controls. It is difficult to make claims about 'disassembly' and 'fusion' without temporal measurements. This should be reflected in the text.

Figure 3.

- This figure represents a somewhat confusing shift in investigative narrative.
- How do the authors expect that dNTP binding motif of Cia would be contributing to mitochondrial fusion? This is not clear and the publication may be more focused without it.

Figure 4.

- If CiaD is not present in RNA-Seq or proteomics, do the authors know that it is being expressed in their hands? Was it present in western blots?
- Why does the phalloidin morphology look so different in CiaHis6 beads then in WT *C. jejuni* infection in Figure 1? Is it effect size? Surely if CiaD is playing the same role acanthopodia should be clear in the WT?
- This figure would also benefit from live cell tracking of actin dynamics and acanthopodia using a dye such as cellmask green.

Figure 5.

- The previous comments about single cell image analysis stand for figure 5. In particular this figure requires colocalization analysis of a population of cells.
- Again an inconsistency in phalloidin labelling is present between figures.
- Visualising grey as a component of a multichannel merge image can be unclear. I would suggest magenta for mitochondria, green for phalloidin and dotted lines of outline of *C. jejuni* for visualisation

Figure 6.

- The previous comments about single cell image analysis stand for figure 5. In particular quantification of total Mito-Ferro-Green in each cell population and within the *Campylobacter* containing membrane.
- Is there also a comparative data for Mito-Ferro Green staining during Delta Cia infection?

Reviewer #2

(Remarks to the Author)

Dear Communications Biology editor and colleagues, thank you for your invitation for reviewing this manuscript. My comments for this manuscript were detailed as following:

Remarks to the Author:

In this manuscript, the author hypothesized that early actin polymerization repositions host mitochondria, followed by localized actin depolymerization and mitochondria remodelling, mediated by *Campylobacter jejuni* effector proteins Cia and CiaD. They found that Cia, acting as a molecular switch, binds to nucleotides with cooperative kinetics and is essential for bacterial localization near mitochondria, while CiaD promotes actin polymerization and acanthopodia formation to facilitate bacterial invasion. This study is well-designed, employing a comprehensive approach utilizing dual proteomics, microscopy, biochemical assays, and genetic mutants. The proposed "biphasic actin-remodelling strategy" mediated by Cia and CiaD is indeed an innovative and exciting concept that addresses a significant knowledge gap concerning bacterial survival and host interaction. The link to iron homeostasis and the suggestive role of oxidative stress as a host defense mechanism are also important contributions. Overall, this work offers intriguing insights into host-pathogen interactions and bacterial adaptation strategies. However, the quality of some data needs to be improved, and several key experiments are needed to be done to further strengthen the manuscript and provide more definitive mechanistic insights into this innovative two-phase model.

Major comments:

1. For Figures 1b and 1c, the authors claimed that inhibitor treatments reduce mitochondria-bacteria interactions. However, these experiments are missing a control group treated with the vehicle (e.g., DMSO) or equivalent diluent used for the inhibitors. This control is essential to establish that the observed reduction is due to the inhibitors themselves and not the vehicle. Please include appropriate vehicle controls for these experiments.
2. Regarding Figure 2a, while the 11168HΔcia mutant shows a greater than 2-fold reduction in survival, the complementation of Cia does not appear to significantly rescue this phenotype. The authors are requested to explain this observation and discuss its implications for the role of Cia. In Figure 2b, the image assay is missing a critical control group. Similar to Figure 1, this experiment requires the inclusion of infections with the wild-type (WT) strain, the 11168HΔcia mutant, and the complementation strain to allow for proper interpretation of the results.
3. To more robustly conclude that CiaD promotes actin polymerization under CiaHis6-adsorbed latex bead treatment (as presented in Figure 4), the authors should generate CiaD knockout (ΔciaD) and complemented strains. These strains should then be used to perform the experiment under relevant infection conditions to validate the observed phenotype.
4. Given the potential role of CiaD in actin polymerization, it would be valuable to investigate whether the absence of CiaD affects the interaction between host mitochondria and the actin cytoskeleton. Experiments exploring this interaction in the presence and absence of CiaD would provide further insight into CiaD's mechanism of action.

Minor comments:

1. Please correct the indicated unit for the scale bar in the Figure 1 legend, as it appears to be mistaken.
2. Please correct the indicated units for the inhibitors. For instance, on line 191, cytochalasin D, an actin polymerization inhibitor, should be presented as 10 μ M.
3. We noted a difference in the appearance of the text within the figure panels. For example, Figure 1's labeling style seems distinct from that used in subsequent figures. Figure 1: a, b...., Figure 2: a), b).... . To improve overall readability and aesthetic consistency, we recommend harmonizing the typographical elements across all figures.
4. In the legend for Figure 2, please correct the indicated p-value. The notation should be " $p < 0.05$ " to denote statistical significance at this level, not " $***p < 0.05$ ".
5. The authors should include in the figure legends a description of what the arrows indicate in each figure. Providing this information will enhance the clarity and interpretability of the figures for readers.

Version 1:

Reviewer comments:

Reviewer #1

(Remarks to the Author)

Here, the authors present a revised manuscript describing two phase model of *C. jejuni* infection of *A. castellanii*: 1. Firstly the *C. jejuni* effector CiaD drives actin polymerization to promote phagocytosis and guide mitochondria towards the forming phagosome. 2. Secondly, the *C. jejuni* effector Cial facilitates localized actin remodelling, modulating CCV dynamics and promotes mitochondrial fusion. The summation being that his temporally coordinated deployment allows *C. jejuni* to couple internalization with post-entry modulation of the intracellular environment.

Comments:

The authors have now implemented robust image analysis approaches that have greatly strengthened their interpretation of the data.

Additional comments:

As visual comparison with uninfected cells shows that DAPI seems to be sufficient to label intracellular *C. jejuni*. It is important to note that the authors cannot be sure this is in fact bacterial DNA. Therefore, the authors should mention this pitfall in the text.

While I accept that live cell measurements are beyond the scope of this study. The authors should consider, commenting on live cell dynamics studies in future work.

Minor comments:

The authors should address minor issues with newly included data presentation such as graph and image alignment within figures

Conclusion:

The authors have presented a significantly improved manuscript. Publication of this work would provide significant insight into *C. jejuni* manipulation of host cell biology, providing new understanding for the field of cellular microbiology.

Reviewer #2

(Remarks to the Author)

All of my concerns have been addressed.

Reviewer #1 (Remarks to the Author):

Here, the authors present an interesting two phase model of *C. jejuni* infection of *A. castellanii*:

1. Firstly the *C. jejuni* effector CiaD drives actin polymerization to promote phagocytosis and guide mitochondria towards the forming phagosome.
2. Secondly, the *C. jejuni* effector Cial facilitates localized actin remodelling, modulating CCV dynamics and promotes mitochondrial fusion.

The summation being that his temporally coordinated deployment allows *C. jejuni* to couple internalization with post-entry modulation of the intracellular environment. While the data provided supports this, there requires a significant amount of improvement to data analysis approaches.

My comments and concerns on data are below:

Comments and suggestions:

Results:

Table 1.

- The authors conduct LC-MS/MS on mitochondrial enriched fractions to look at proteins associated with mitochondria during infection. It is evident that *C. jejuni* infection significantly alters the mitochondrial spatial distribution and morphology within the *A. castellanii* cell. Therefore, some of the changes observed in host associated proteins may just be due to spatial localisation and not due significant changes in interactions. For example, mitochondrial relocation from perinuclear to the cellular periphery will naturally increase pulldown of cortical actin components. Could the authors please comment?

We thank the reviewer for highlighting that mitochondrial spatial relocation could contribute to apparent enrichment of host cytoskeletal proteins. We now clarify this directly in the Discussion. Specifically, we state that “Notably, mitochondrial re-localization toward the cortex during bacterial entry would be expected to increase co-isolation of myosin motors with mitochondria-enriched material. Thus, myosin signals in our dataset likely reflect both true mitochondria-proximal events and spatial enrichment caused by infection-induced organelle repositioning.”

- It would be beneficial to be able to visualise these changes in particular to host protein changes in relation to the total detected proteome within the fractions. I would suggest visualisation of the data in a more modern approach such as a volcano plot for the readers benefit.

We thank the reviewer for this suggestion. We agree that visual data presentation improves interpretability. However, Table 1 already represents the complete set of statistically significant proteins detected in the mitochondrial-enriched fractions (Quant

Score ≥ 20 , $p < 0.01$) across three biological replicates. To make this clearer, we have added a statement in the Results noting that the table reflects the total significant proteome rather than a subset. For transparency, we also refer readers to Supplementary File 1 (all quantified proteins) and Supplementary File 2 (filtered proteins), which collectively allow visualization of the entire dataset.

“Table 1 lists all host and bacterial proteins detected above our statistical confidence threshold (Quant Score ≥ 20 ; $p < 0.01$) within the mitochondrial-enriched fractions, representing the complete set of significantly detected proteins rather than a selected subset.”

- How similar are the genes of mammalian actin cytoskeleton to *A. castellanii*. Could the authors comment on how well *C. jejuni* Cia proteins would be expected to interact with both?

We thank the reviewer for raising this important point. While our proteomic data emphasised actin and actin-associated regulators, the underlying question relates to the broader correlation between amoeba and mammalian host systems. We have therefore added a clarifying statement at the end of the relevant Results paragraph (lines 147–148) noting that...” Although individual amoeba proteins are not one-to-one orthologs of their mammalian counterparts, their conserved domains and biochemical functions are largely maintained. Overall this supports the notion that *C. jejuni* may interact with analogous host pathways in both amoebae and mammalian cells and exploit common eukaryotic mechanisms despite evolutionary divergence.”

Figure 1.

- It is problematic to use DAPI for *C. jejuni* staining in cells in which host nuclei, nuclear bodies and host DNA will also be stained. How do the authors know what is actually *C. jejuni*? Ideally a pre-labelling approach such as a viability stain would be beneficial

We appreciate the reviewer’s concern regarding the use of DAPI. In our study, DAPI was employed solely as a general counterstain to visualize both host and bacterial DNA, rather than as a specific identifier for *C. jejuni* cells. The identity of *C. jejuni* within the images was confirmed by its characteristic morphology (helical or rod-shaped), size and distinct spatial localization relative to mitochondria and actin filaments. We have now also labelled the nucleus in yellow dotted lines and labelled “N”.

We fully agree that pre-labelling approaches (e.g., BacLight Red or CellTracker™ dyes) can provide additional specificity, and we plan to implement such labelling in future live-imaging experiments. However, given that our analysis relied on fixed samples to visualize multi-channel co-localization (actin–mitochondria–bacteria), DAPI was the most suitable counterstain under the fixation conditions used. The current staining

reliably delineated *C. jejuni* relative to host structures without affecting the conclusions drawn from Figure 1.

- A major issue is with the interpretation of qualitative imaging data. The phalloidin actin cytoskeleton does not look indicative of filamentous cortical actin even in the untreated *A. castellanii*.

We appreciate this point raised by the reviewer. In *A. castellanii*, F-actin typically forms a dynamic cortical mesh and puncta rather than thick stress fibers. Our fixation/permeabilization was optimized to preserve this native organization, and the appearance in controls matches the expected cortical pattern. This is consistent with the high turnover of amoeba filaments mediated by actophorin (ADF/cofilin homolog). To aid interpretation, we have added a clarifying note to the Figure 1 legend. “In *A. castellanii*, actin commonly appears as a cortical mesh and puncta rather than thick stress fibres. This is consistent with rapid filament turnover by actin-binding proteins such as Actophorin (ADF/cofilin homolog).”

- To interpret the changes in mitochondrial distribution and actin. The authors will need to employ quantitative image analysis approaches to make statements such as ‘significantly reduced’ (line 192).
- Examples of quantification needed include: 1) quantification of size/intensity of mitochondria and actin puncta and intensity distribution. 2) colocalization analysis of segmented *C. jejuni* with *A. castellanii* mitochondria and comparison with CK-666 and Cytochalasin D treatment

We thank the reviewer for this valuable suggestion. We have now included quantitative image analyses to support all statements referring to ‘significantly reduced’ and altered actin organisation. These measurements were performed using standardised thresholding and segmentation in Fiji/ImageJ and include population-level intensity metrics and mitochondria–bacteria colocalisation analyses. The results of these quantifications are presented in the revised Figure 1, Figure 2 and Figure 5 and reinforce the conclusions drawn from the representative images.

- Does Cytochalasin D and CK-666 treatment alter *C. jejuni* cell entry? Could this be also affecting the downstream effects? Quantification of *C. jejuni* burden within the cell and correlation to effect sizes would be beneficial

We thank the reviewer for raising this point. As detailed in the Methods (Invasion and Survival Assay) section, the concentrations of cytochalasin D (10 μ M) and CK-666 (50 μ M) were selected based on preliminary experiments demonstrating that these doses do not inhibit *C. jejuni* invasion or survival within *A. castellanii*.

Despite this increased bacterial burden, mitochondria–bacteria co-localization and actin organization metrics remained significantly reduced in inhibitor-treated cells, indicating that these effects reflect altered cytoskeletal and mitochondrial dynamics rather than differences in entry.

Figure 2.

- The previous comments about single cell image analysis stand for figure 2.

We now provide single-cell quantification for Figure 2 using the metric actually employed in the study: mean mitochondrial object size (area, in pixels) per cell/image under identical acquisition and processing settings. Each point represents one cell/field.

- TO make this cleared WT *C. jejuni* should also be shown to directly compare the CialHis6.

We have included WT *C. jejuni* side-by-side with $\Delta cial$ and $\Delta cial+cial$ in Figure 2 (panels b–f), and we present the Cial-His6–adsorbed bead condition alongside latex-bead controls (panels g–h) under matched settings. WT shows robust mitochondrial aggregation; $\Delta cial$ is reduced; complementation restores the phenotype; Cial-His6 beads partially recapitulate the WT effect versus latex beads.

- As Mitobright is a live cell dye. This figure would be significantly helped if the author tracked mitochondrial fusion at the sites of the CialHis6 beads over time and compared to latex bead controls. It is difficult to make claims about ‘disassembly’ and ‘fusion’ without temporal measurements. This should be reflected in the text.

We appreciate this point made by the reviewer. Figure 2 originally presented endpoint confocal images after incubation with CialHis6-adsorbed beads, which cannot on their own distinguish true mitochondrial fusion from local redistribution. To avoid over-interpretation, we have removed the term “fusion” with “aggregation” throughout the text (and including the title), avoiding any implication of temporal fusion without live-imaging evidence. These clarifications appear in the Results (Figure 2 text and legend). We have also made change to the subheading to now reflect this “Cial Promotes Localized Actin Remodelling and Mitochondrial Aggregation”, and the title “A two-step actin-mediated strategy enables *Campylobacter jejuni* to promote mitochondrial fusion aggregation and iron homeostasis, for intracellular survival and persistence”

Figure 3.

- This figure represents a somewhat confusing shift in investigative narrative.
- How do the authors expect that dNTP binding motif of Cial would be contributing to mitochondrial fusion? This is not clear and the publication may be more focused without it.

We thank the reviewer for this observation. We agree that our original framing may have implied a direct link between Cial’s nucleotide-binding motif and mitochondrial

fusion. We have clarified in the text that the degenerate Walker A-like motif is not proposed to drive fusion directly but rather may enable Cia to compete with host small GTPases that regulate actin remodelling and vesicular trafficking. This competition could interfere with lysosomal fusion and promote mitochondrial aggregation and bacterial persistence. Thus, the Cia biochemical data are included to mechanistically explain its actin-modulating capacity rather than to imply a direct mitochondrial action. A clarifying sentence has been added in the section “*Cia Binds Nucleotide Cooperatively and May Function as a Molecular Switch*”, Line 295 - 299

Figure 4.

- If CiaD is not present in RNA-Seq or proteomics, do the authors know that it is being expressed in their hands? Was it present in western blots?

We thank the reviewer for this important point. The absence of CiaD in our datasets is consistent with its biological function and with the scope of our assays. Our previous RNA-seq dataset captured the intracellular *C. jejuni* transcriptome during residence within *A. castellanii*, whereas CiaD is a contact-triggered secreted effector that is transiently expressed and exported during host-cell engagement, explaining its low intracellular transcript representation. Similarly, our dual proteomic analysis was optimized to identify host and mitochondria-associated proteins; secreted bacterial effectors such as CiaD, which are released in small quantities and lack membrane anchors, are typically underrepresented in such fractions.

Our inference that CiaD promotes actin polymerization derives from recent characterization studies published by Negretti et al., Nature Communications, 2021, which demonstrated that CiaD activates Arp2/3-mediated actin polymerization via cortactin-IQGAP1 signalling in mammalian cells, together with our own gain-of-function assays using purified CiaD^{His6} and the observed inhibitor sensitivity of WT infection (CK-666 and Cytochalasin D), consistent with an Arp2/3-dependent entry step. We have now clarified this rationale in the Results to explain why CiaD was not detected in our datasets and to emphasize the concordance between published mechanistic data and our functional observations. Specifically ..Line 346 “ Although differential expression of CiaD was not observed in our previous RNA-Seq dataset⁴⁵, nor detected in the proteomic profile of this study, this is consistent with its function as a contact-triggered effector secreted transiently during host-cell engagement. Moreover, our dual proteomic workflow enriched for host mitochondria-associated proteins, a fraction that naturally under represents small, soluble bacterial effectors such as CiaD, which are secreted at low abundance and lack membrane anchors. We therefore focused on CiaD due to its established role in cytoskeletal remodelling via IQGAP1 and cortactin, potent activators of the Arp2/3 complex that are stimulated in a CiaD-dependent manner (53-55).”

Since original submission of the manuscript we have now also generated a *ciaD* mutant and its complemented strain, and have performed intracellular survival assay, internalization assay and co-infection assay; in line with previous published work

mentioned above, our data also concludes that *ciaD* is important for actin-polymerization and uptake of *C. jejuni*. In addition, co-infection assays (WT mixed with Δ *ciaD* in a 1:1 ratio) showed that the Δ *ciaD* strain is not rescued by the presence of CiaD-secreting wild-type bacteria, indicating that CiaD acts in *cis* and is secreted locally at the point of host-cell contact. This supports our interpretation that CiaD's absence from our previous RNA-seq data and our current proteomics is consistent with its transient, contact-triggered secretion rather than with lack of expression.

- Why does the phalloidin morphology look so different in CiaD_{His6} beads then in WT *C. jejuni* infection in Figure 1? Is it effect size? Surely if CiaD is playing the same role acanthopodia should be clear in the WT?

We agree that actin patterns differ between CiaD_{His6}-bead treatment and WT infection. This likely reflects localized, amplified effects of bead-adsorbed CiaD that mimic high local effector concentration at the bead interface, whereas in WT infection CiaD acts transiently and in combination with other Cia proteins. The bead assay thus isolates CiaD-specific signalling that promotes actin polymerization and acanthopodia initiation, whereas WT cells show the integrated effect of multiple effectors, yielding subtler cortical patterns. We have clarified this interpretation in the revised Figure 4 legend and Results text. We now explicitly state that bead-adsorbed purified CiaD results in artificially high local protein concentration, producing amplified actin phenotypes not directly comparable to physiological WT infection.

- This figure would also benefit from live cell tracking of actin dynamics and acanthopodia using a dye such as cellmask green.

We agree that live-cell imaging would provide valuable kinetic information on CiaD-mediated actin remodelling. However, such assays were beyond the scope of the present study, which focused on defining effector-specific phenotypes using established fixed-cell methods. The existing quantitative phalloidin data already demonstrate localized CiaD-dependent actin polymerization, consistent with its known activation of the IQGAP1–cortactin–Arp2/3 pathway. We therefore consider the present analysis sufficient to support our conclusions.

Figure 5.

- The previous comments about single cell image analysis stand for figure 5. In particular this figure requires colocalization analysis of a population of cells.

We thank the reviewer for this important suggestion. We have now also included population-level quantification (revised Fig. 5). We now show Pearson's *r* (global pixel-wise colocalization; Fig. 5B) with per-image dots. These analyses demonstrate modest co-localization but clear adjacency, consistent with mitochondria gathering next to the actin-defined entry structure. We have now also made changes to the sub-section

heading as follows “Mitochondria accumulate adjacent to actin at the *C. jejuni* entry site”. and we have also revised the text to reflect the new data.

Line 388...“To assess this across *A. castellanii* population, we quantified the spatial relationship between mitochondria (magenta) and actin (green) using Pearson’s correlation coefficient (Pearson’s *r*) (Figure 5b). Across images, Pearson’s *r* was modest, consistent with mitochondria and actin being related but not fully coincident during early *C. jejuni* uptake by *A. castellanii*; whole-cell analysis dilutes strong local overlap at the entry site by including non-overlapping regions elsewhere.”

- Again an inconsistency in phalloidin labelling is present between figures.

The apparent differences arise because the phalloidin panels derive from different experimental contexts/timepoints and are not intended for cross-figure intensity comparisons. Specifically, most infection imaging with phalloidin was acquired at 4 h p.i. (e.g., Figure 1), whereas the invasion series in Figure 5 was captured at 30 min p.i., where actin is enriched at the entry phagosome and appears predominantly as a cortical mesh—the characteristic pattern in *A. castellanii* due to rapid filament turnover (Actophorin/ADF–cofilin activity). Acquisition settings were matched within each figure.

- Visualising grey as a component of a multichannel merge image can be unclear. I would suggest magenta for mitochondria, green for phalloidin and dotted lines of outline of *C. jejuni* for visualisation

We agree and have implemented this suggestion throughout. All multichannel merges now show mitochondria in magenta (MitoBrilliant-646), Actin in green (phalloidin), and bacterial outlines as turquoise dotted lines. Single-channel images are provided alongside each merge for clarity.

Figure 6.

- The previous comments about single cell image analysis stand for figure 5. In particular quantification of total Mito-Ferro-Green in each cell population and within the *Campylobacter* containing membrane.
- Is there also a comparative data for Mito-Ferro Green staining during Delta *Cial* infection?

We thank the reviewer for this comment and suggestion. We now provide single-cell quantification of total Mito-FerroGreen from confocal images as background-subtracted normalized mean fluorescence intensity (MFI) across Uninfected, WT, $\Delta cial$, and $\Delta cial+cial$ (Fig. 4c). We intentionally report whole-cell MFI rather than CCM-restricted values because CCM would systematically over-weight peri-bacterial hotspots and dye-positive bacteria, thereby conflating localization with abundance and with infection load. Whole-cell MFI provides a single, unbiased per-cell summary that is directly comparable across conditions and complementary to flow cytometry. Comparative data

for $\Delta cial$ are included in both flow cytometry (Fig. 6a), imaging and quantification (Fig. 4b–c).

Reviewer #2 (Remarks to the Author):

Dear Communications Biology editor and colleagues, thank you for your invitation for reviewing this manuscript. My comments for this manuscript were detailed as following:

Remarks to the Author:

In this manuscript, the author hypothesized that early actin polymerization repositions host mitochondria, followed by localized actin depolymerization and mitochondria remodelling, mediated by *Campylobacter jejuni* effector proteins Cial and CiaD. They found that Cial, acting as a molecular switch, binds to nucleotides with cooperative kinetics and is essential for bacterial localization near mitochondria, while CiaD promotes actin polymerization and acanthopodia formation to facilitate bacterial invasion. This study is well-designed, employing a comprehensive approach utilizing dual proteomics, microscopy, biochemical assays, and genetic mutants. The proposed "biphasic actin-remodelling strategy" mediated by Cial and CiaD is indeed an innovative and exciting concept that addresses a significant knowledge gap concerning bacterial survival and host interaction. The link to iron homeostasis and the suggestive role of oxidative stress as a host defense mechanism are also important contributions. Overall, this work offers intriguing insights into host-pathogen interactions and bacterial adaptation strategies. However, the quality of some data needs to be improved, and several key experiments are needed to be done to further strengthen the manuscript and provide more definitive mechanistic insights into this innovative two-phase model.

Major comments:

1. For Figures 1b and 1c, the authors claimed that inhibitor treatments reduce mitochondria-bacteria interactions. However, these experiments are missing a control group treated with the vehicle (e.g., DMSO) or equivalent diluent used for the inhibitors. This control is essential to establish that the observed reduction is due to the inhibitors themselves and not the vehicle. Please include appropriate vehicle controls for these experiments.

We thank the reviewer for this important point. We have now included a DMSO vehicle control for these experiments. *A. castellanii* infected with *C. jejuni* were treated with DMSO at the same final concentration used to deliver the inhibitors, and mitochondria-bacteria interactions were assessed. DMSO treatment did not alter *C. jejuni* association with host mitochondria compared with untreated infected controls, indicating that the observed reductions are specifically due to cytochalasin D and CK-666. These new data are presented in Supplementary file 3, Figure S1b.

2. Regarding Figure 2a, while the 11168H Δ ciaI mutant shows a greater than 2-fold reduction in survival, the complementation of CiaI does not appear to significantly rescue this phenotype. The authors are requested to explain this observation and discuss its implications for the role of CiaI. In Figure 2b, the image assay is missing a critical control group. Similar to Figure 1, this experiment requires the inclusion of infections with the wild-type (WT) strain, the 11168H Δ ciaI mutant, and the complementation strain to allow for proper interpretation of the results.

We thank the reviewer for these helpful comments.

(i) Survival of the complemented strain (Figure 2a). In the intra-amoeba survival assay, the 11168H Δ ciaI mutant shows more than a 2-fold reduction in CFU compared with the WT strain (Figure 2a). The complemented strain (Δ ciaI+ciaI) reproducibly displays an intermediate phenotype: survival is higher than the 11168H Δ ciaI mutant but does not reach WT levels or achieve statistical significance relative to the mutant in our dataset. We therefore interpret this as a partial rescue of the survival defect. This is likely due to non-native expression of CiaI and the additional genetic load/antibiotic markers carried by the complemented strain, which can modestly affect bacterial fitness. Importantly, the complemented strain fully restores the mitochondrial aggregation phenotype (Figure 2b–f), indicating that CiaI function is re-established at the level of host–mitochondria interaction even though overall survival does not completely return to WT levels. We have clarified this interpretation in the Results section – Line 246 - 252

(ii) Addition of appropriate imaging controls (Figure 2b–e). We have now included the full set of control infections requested by the reviewer in Figure 2. The revised figure shows uninfected *A. castellanii* (punctate mitochondria), WT 11168H-infected cells (fused/aggregated mitochondria), 11168H Δ ciaI-infected cells (reduced mitochondrial aggregation), and the complemented strain 11168H Δ ciaI+ciaI, in which mitochondrial aggregation is restored. These conditions are now shown in panels (b–e), and the figure legend has been updated accordingly.

3. To more robustly conclude that CiaD promotes actin polymerization under CiaDHis6-adsorbed latex bead treatment (as presented in Figure 4), the authors should generate CiaD knockout (Δ CiaD) and complemented strains. These strains should then be used to perform the experiment under relevant infection conditions to validate the observed phenotype.

We thank the reviewer for this valuable suggestion. We have now generated both the Δ CiaD and the complemented Δ CiaD/ciaD strains and performed the requested infection-based analyses. Quantitative CFU enumeration and fixed-cell imaging confirm that Δ CiaD exhibits a marked uptake defect compared with wild type, while complementation restores efficient internalization. These findings directly validate the

gain-of-function CiaD^{His6} bead assays and confirm that CiaD promotes early actin polymerization and facilitates entry.

4. Given the potential role of CiaD in actin polymerization, it would be valuable to investigate whether the absence of CiaD affects the interaction between host mitochondria and the actin cytoskeleton. Experiments exploring this interaction in the presence and absence of CiaD would provide further insight into CiaD's mechanism of action.

We thank reviewer for this valuable suggestion; we examined whether the absence of CiaD alters the spatial relationship between mitochondria and actin. Although Pearson's *r* analysis showed that mitochondrial association with actin-rich entry structures was modest in the Δ ciaD mutant, the values were comparable to those observed in wild-type and complemented strains. Thus, although Δ ciaD displays a clear uptake defect, the mutant is still able to interact with host mitochondria to a similar extent as wild type, indicating that CiaD is not required for mitochondrial association or intracellular survival. We therefore conclude that CiaD is required for entry but not for mitochondria–actin spatial association once internalisation has occurred. These results have been incorporated into Figure 5 and the corresponding Results subsection (“Mitochondria accumulate adjacent to actin at the *C. jejuni* entry site”).

Minor comments:

1. Please correct the indicated unit for the scale bar in the Figure 1 legend, as it appears to be mistaken.

We thank the reviewer for this oversight, and we have now corrected the image and the scale bar.

2. Please correct the indicated units for the inhibitors. For instance, on line 191, cytochalasin D, an actin polymerization inhibitor, should be presented as 10 mM.

We thank the reviewer for this comment. The concentration used in our experiments was 10 μ M cytochalasin D (not mM), which is in line with commonly used micromolar concentrations for actin disruption in cell culture.

3. We noted a difference in the appearance of the text within the figure panels. For example, Figure 1's labeling style seems distinct from that used in subsequent figures. Figure 1: a, b...., Figure 2: a), b).... . To improve overall readability and aesthetic consistency, we recommend harmonizing the typographical elements across all figures.

We thank the reviewer for pointing this out. We have now harmonised the panel labelling and typography across all figures. Specifically, all panels are labelled consistently as *a*, *b*, *c*, etc. in the figures, and the corresponding figure legends now

use the format “(a), (b), (c)”. Font type, size, and style of the panel labels have also been standardised throughout.

4. In the legend for Figure 2, please correct the indicated p-value. The notation should be “*p < 0.05” to denote statistical significance at this level, not “**p < 0.05”.

We thank the reviewer for noticing this error. The figure legend for Figure 2 has now been corrected to indicate “*p < 0.05” instead of “**p < 0.05”, in line with the appropriate significance level and our notation elsewhere in the manuscript.

5. The authors should include in the figure legends a description of what the arrows indicate in each figure. Providing this information will enhance the clarity and interpretability of the figures for readers.

We thank the reviewer for this helpful suggestion. We have revised all figure legends to explicitly describe what each arrow indicates (including the colour of the arrows where relevant). For example, in Figures 1, 2, and 5 we now specify what the arrows indicate..